# Identifiable Interpretation in Generative Models via Causal Minimality

## Abstract

Deep generative models, while revolutionizing fields like image and text generation, largely operate as opaque "black boxes", hindering human understanding, control, and alignment. While methods like sparse autoencoders (SAEs) show remarkable empirical success, they often lack theoretical guarantees, risking subjective insights. Our primary objective is to establish a principled foundation for interpretable generative models. We demonstrate that the principle of causal minimality – favoring the simplest causal explanation – can endow the latent representations of diffusion vision and autoregressive language models with clear causal interpretation and robust, component-wise identifiable control. We introduce a novel theoretical framework for hierarchical selection models, where higher-level concepts emerge from the constrained composition of lower-level variables, better capturing the complex dependencies in data generation. Under theoretically derived minimality conditions (manifesting as sparsity or compression constraints), we show that learned representations can be equivalent to the true latent variables of the data-generating process. Empirically, applying these constraints to leading generative models allows us to extract their innate hierarchical concept graphs, offering fresh insights into their internal knowledge organization. Furthermore, these causally grounded concepts serve as levers for fine-grained model steering, paving the way for transparent, reliable systems.

## 1 Introduction

The transformative power of deep generative models, including diffusion models (Sohl-Dickstein et al., 2015; Ho et al., 2020; Rombach et al., 2022b; Song et al., 2022; Dhariwal & Nichol, 2021; Nichol & Dhariwal, 2021) and language models (Radford et al., 2018; 2019; Brown et al., 2020; Raffel et al., 2020), is reshaping numerous domains. However, their escalating complexity and scale frequently cast them as opaque "black boxes" (Shwartz-Ziv & Tishby, 2017; Olah et al., 2020). This opacity presents a formidable barrier to genuine human understanding, severely curtails our ability to exert precise control over their behavior (Jahanian et al., 2019; Härkönen et al., 2020; Shen et al., 2020a; Wu et al., 2021), and complicates the crucial alignment with human values and intentions.

Although recent empirical tools, such as sparse autoencoders (SAEs) for large language models (LLMs) (Cunningham et al., 2023; Huben et al., 2023; Gao et al., 2024) and diffusion models (Surkov et al., 2024; Kim et al., 2024; Kim & Ghadiyaram, 2025; Cywiński & Deja, 2025; Huang et al., 2025), offer avenues for probing these models, a fundamental gap persists. Without rigorous theoretical underpinnings, interpretations derived from these methods risk being subjective or susceptible to human biases, rendering them potentially untrustworthy for risk-sensitive applications (Kaddour et al., 2022; Moran & Aragam, 2025; Schölkopf et al., 2021). In this work, we directly tackle this critical challenge, seeking to establish a principled foundation for interpretable and controllable generative models.

Our investigation centers on two questions: Under what *theoretical conditions* can we reliably identify meaningful, interpretable latent concepts within the intricate architectures of modern generative models? And, crucially, what *actionable, theoretically-grounded insights* can empower us to advance both the interpretability and the controllability of these powerful systems?

Towards these goals, we identify the *causal minimality* (Peters et al., 2017; Spirtes et al., 2000; Hitchcock, 2021) principle as the formal underpinning that connects widespread practices, such as

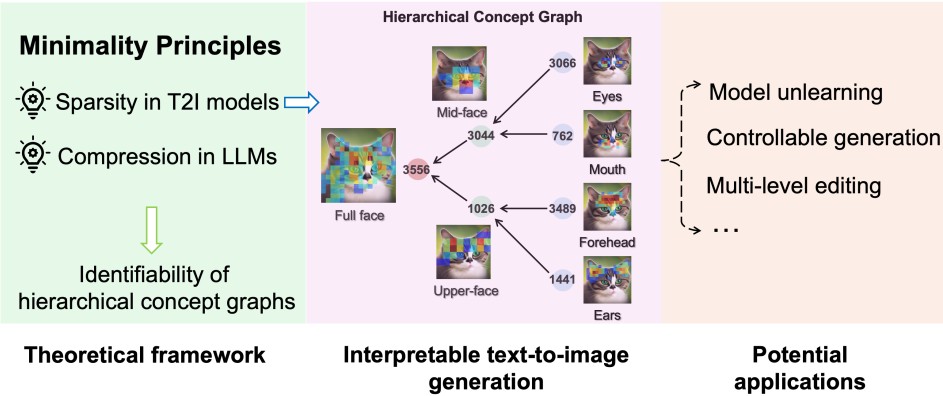

Figure 1: Our causal minimality principle enables interpretable text-to-image generation through hierarchical concept graphs, with implications for downstream tasks.

enforcing sparsity, to the recovery of meaningful, interpretable concepts. This principle, advocating for the simplest causal model consistent with observations, allows for the identification of latent hierarchical concept structures. In our context, minimality translates to either sparsity in the concept graphs or the most compressed active discrete concept states. We explore its application to text-to-image (T2I) diffusion models (Ramesh et al., 2021; 2022; Rombach et al., 2022b) and autoregressive language models (LMs) (Radford et al., 2018; 2019; Brown et al., 2020). Our findings indicate that imposing sparsity constraints on internal representations is instrumental for identifying intrinsic visual concepts and textual concepts.

A cornerstone of our contribution is establishing the first identifiability results for *selection*-based (Zheng et al., 2024; Spirtes et al., 1995; Hernán et al., 2004; Zhang, 2008; Bareinboim et al., 2022; Forré & Mooij, 2020; Correa et al., 2019; Chen et al., 2024) hierarchical models. In such models, higher-level variables emerge as effects of compositions of lower-level variables, where higher-level variables control and select the configuration of lower-level ones. This fundamentally diverges from traditional hierarchical causal models (Pearl, 2009; Choi et al., 2011; Zhang, 2004), in which causal influence typically propagates from higher to lower levels. The selection model structure is particularly adept at capturing the intricate conditional dependencies among low-level features for forming coherent high-level concepts – it explains how specific arrangements of wheels, doors, and a roof constitute a recognizable "car", rather than a disjoint collection of parts. Traditional hierarchical models often neglect such intra-level dependencies by assuming no within-layer causal edges, as explicitly modeling them would yield overly dense graphs. The selection mechanism, in contrast, offers a simpler approach to this essential coordination. Its adherence to the minimality principle strongly favors it as a more accurate representation of the true model.

Despite the appeal, their identifiability has been underexplored. Prior research has largely centered on traditional hierarchical structures. Moreover, their techniques often rest on simplifying assumptions (e.g., linearity (Xie et al., 2022; Huang et al., 2022; Dong et al., 2023; Anandkumar et al., 2013) or achieve only subspace-level identifiability (Kong et al., 2023a)). Such methods are generally inapplicable to the hierarchical selection models. Our framework is the first to establish *component-wise* identifiability for both *continuous and discrete hierarchical selection* models. Specifically, we demonstrate that under well-defined minimality conditions (Conditions 4.2-iv and A.1-iii), the learned representations are equivalent to the true latent variables of the underlying hierarchical process. This disentanglement of individual, atomic concepts is what affords significantly more nuanced interpretability and precise control in the resulting generative models.

By applying the derived sparsity constraints to state-of-the-art generative models, we successfully extract their innate hierarchical concept graphs (Figure 1). This not only illuminates their internal knowledge organization but also shows that causally-grounded concepts serve as highly effective levers for model steering. Our experiments illustrate key implications of our theorems and show how a principled, causal understanding can guide the application of established interpretation techniques.

Due to the page limit, we focus on the visual concept identification with T2I models in the main text and defer the text counterpart with LMs to Appendix A.

## 2 RELATED WORK

**Hierarchical models.** Complex real-world data distributions frequently exhibit inherent hierarchical structures among their underlying latent variables, a characteristic that has motivated extensive research. Initial explorations primarily focus on continuous latent variables with linear interactions (Xie et al., 2022; Huang et al., 2022; Dong et al., 2023; Anandkumar et al., 2013). Other lines of work have centered on discrete latent variables; however, these approaches are often constrained in their applicability to continuous data modalities like images (Pearl, 1988; Zhang, 2004; Choi et al., 2011; Gu & Dunson, 2023; Kong et al., 2024). Furthermore, prevalent latent tree models, which connect variables via a single undirected path (Pearl, 1988; Zhang, 2004; Choi et al., 2011), risk oversimplifying the multifaceted relationships present in complex systems. More recently, while Park et al. (2024) make progress in capturing geometric properties of language model representations using hierarchical models, their work does not address the critical issue of latent variable identification. Kong et al. (2023a) tackle nonlinear, continuous latent hierarchical models, but their framework, operating under rather opaque functional conditions, falls short of component-wise identifiability, thereby leaving room for concept entanglement. Our work distinctively investigates *selection* hierarchical models, contending that their structural properties yield a more faithful representation of latent concepts in natural data distributions. In these models, latent variables function as colliders, a significant departure from their role as confounders in the aforementioned prior art. This critical distinction renders existing identification techniques largely inapplicable. To the best of our knowledge, we are the first to provide *component-wise* identifiability for *both continuous and discrete hierarchical selection models*.

**Interpretability for generative models.** Despite the remarkable advancements of generative models, their internal mechanisms often remain opaque. This presents a significant challenge to understanding and control. Considerable research has focused on obtaining interpretable features to enable more controllable generation. Early efforts center on analyzing the latent space of generative adversarial networks, e.g., (Härkönen et al., 2020; Voynov & Babenko, 2020; Shen et al., 2020b). Recently, sparse autoencoders (SAEs) have gained prominence for interpreting hidden representations, particularly in language models. These studies show that SAEs trained on transformer residual-stream activations can identify latent units corresponding to linguistically meaningful features (Cunningham et al., 2023; Huben et al., 2023; Gao et al., 2024; Mudide et al., 2025; Shi et al., 2025). These interpretability techniques have also been successfully extended to diffusion models. Surkov et al. (2024) reveal interpretable features and specialization across diffusion model blocks. Other work trains SAEs with lightweight classifiers on diffusion model features (Kim et al., 2024) or steers generation away from undesirable visual attributes (Huang et al., 2025). Our hierarchical approach is related to recent findings on the evolution of semantics during the diffusion process. It has been observed that high-level concepts, such as object shape and structure, tend to emerge in earlier, high-noise timesteps, while fine-grained, low-level details are synthesized in later, low-noise stages (Patashnik et al., 2023; Tinaz et al., 2025; Mahajan et al., 2024). While these works provide valuable empirical validation of this phenomenon, our work offers a new perspective by framing these observations within a formal hierarchical, causal structure. We provide a theoretical foundation, rooted in causal minimality and selection models, to explain *how* these concepts compose and, crucially, *under what conditions* they can be provably identified. Our approach also relates to generative concept bottleneck models, which achieve interpretability by forcing predictions through a bottleneck layer of concepts (Ismail et al., 2024; Kulkarni et al., 2025). While these methods provide powerful intervention capabilities by design, our work differs by focusing on the discovery of the innate hierarchical and causal concept structure in the data. We provide the theoretical conditions for identifying these concepts component-wise, allowing us to then use this discovered graph for fine-grained multi-level interventions.

**Decomposition-based interpretability.** Our work is fundamentally distinct from post-hoc, decomposition-based interpretability methods, such as the prototype-matching approach (Chen et al., 2019). This line of research, while pioneering, has known limitations (often stemming from its prototype-based implementation): its reliance on class-label supervision can lead to non-compositional, class-locked concepts (Rymarczyk et al., 2021), and its use of rigid patch-matching struggles with context and deformation (Donnelly et al., 2022; Xue et al., 2024). In contrast, our approach is *class/object agnostic* (similar to SAEs) and *context-sensitive*, learning from the raw generative data without class labels. Our approach learns compositional, shared concepts (e.g., a single "furry texture" from "cats" and "pandas") rather than rigid, class-specific prototypes. This

enables the causal, interventional control (e.g., Figure 11 and downstream tasks in Section 5.2) that prototype-matching cannot guarantee.

Please find additional related work in Appendix B.

## 3 DEEP GENERATIVE MODELS AS HIERARCHICAL CONCEPT MODELS

**Notations.** We denote random variables with upper-case characters (e.g., $X$) and values with lower-case characters (e.g., $x$). We distinguish multidimensional objects with bold fonts (e.g., $\mathbf{X}$) and refer to their dimensionality as $n(\cdot)$. We view multidimensional variables as *sets* when appropriate (e.g., $\mathbf{X}$ as $\{X_i\}_{i \in [n(\mathbf{X})]}$). Parents $\mathrm{Pa}(\cdot)$ and children $\mathrm{Ch}(\cdot)$ relations are defined based on the selection graph (Figure 2). If $X$ has only one child $Y$, we refer to $X$ as a pure parent of $Y$, i.e., $X \in \mathrm{PPa}(Y)$; if $X$ has other children than $Y$, we refer to $X$ as a hybrid parent of $Y$, i.e., $X \in \mathrm{HPa}(Y)$. We denote the set of natural numbers $\{1, \ldots, M\}$ as $[M]$. More background information is in Appendix D.

We denote the image as the continuous variable $\mathbf{X} \in \mathbb{R}^{n(\mathbf{X})}$ and text as the discrete variable $\mathbf{D} \in \mathbb{N}^{n(\mathbf{D})}$. Visual concepts are $\mathbf{Z} := [\mathbf{Z}_1, \cdots, \mathbf{Z}_{L_\mathrm{V}}]$, where $L_\mathrm{V}$ is the number of visual hierarchical levels and $\mathbf{Z}_l \in \mathbb{R}^{n(\mathbf{Z}_l)}$ are concepts at level $l$ (Figure 2). The discrete variables $\mathbf{D}$ capture the discrete nature of textual concepts (like "cat" or "bicycle"). In contrast, the visual concepts ($\mathbf{Z}$) are continuous to represent rich visual details. $\mathbf{D}$ acts as a selection variable that governs the joint configuration of the continuous $\mathbf{Z}$ variables. For instance, the discrete concept "bicycle" ($\mathbf{D}$) selects for a coherent arrangement of continuous visual features ($\mathbf{Z}$) representing wheels, a frame, and handlebars, rather than a random collection of those continuous parts.

**Hierarchical processes and selection mechanisms.** Our framework conceptualizes high-level concepts as emerging from or being effects of lower-level concepts. This is captured by a *selection mechanism* (Zheng et al., 2024; Spirtes et al., 1995; Hernán et al., 2004; Zhang, 2008; Bareinboim et al., 2022; Forré & Mooij, 2020; Correa et al., 2019; Chen et al., 2024), where variables $\mathbf{V}_l$ at a higher level of abstraction (smaller $l$) is determined by its constituent, more detailed components $\mathbf{V}_{l+1}$ (i.e., its "parents"). The selection function $g_{\mathbf{V}_l}$ maps these lower-level constituents to the higher-level concept:

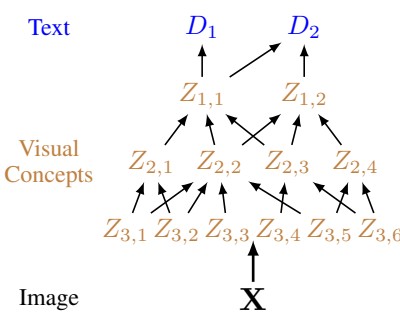

$$\mathbf{V}_l := g_{\mathbf{V}_l}(\mathbf{V}_{l+1}). \tag{1}$$

In other words, $\mathbf{V}_l$ is a selection variable over $\mathbf{V}_{l+1}$. In many natural data distributions of interest, we can only observe the data points for which the selection criterion is met, i.e., $\mathbf{V}_l$ only takes on a strict subset of its range $\Omega$. Therefore, the distribution of $\mathbf{V}_{l+1}$ is always the conditional distribution $\mathbb{P}(\mathbf{V}_{l+1}|\mathbf{V}_l)$. This conditioning on $\mathbf{V}_l$ can induce dependencies among components in $\mathbf{V}_{l+1}$. For instance, if $V_{l+1,i} \to V_l \leftarrow V_{l+1,j}$, conditioning on $V_l$ makes $V_{l+1,i}$ and $V_{l+1,j}$ dependent.

Figure 2: **A visual concept graph.** We denote text as $\mathbf{D}$, visual concepts as $\mathbf{Z}$, and the image as $\mathbf{X}$. High-level concepts function as selection variables for low-level variables. See Figure 7 for the text counterpart.

Under this formulation, one can leverage the inverse process of (1) to sample observable data (images, text), proceeding from higher-level abstract concepts to lower-level concrete details:

$$\mathbf{Z}_0 \sim \mathbb{P}(\mathbf{Z}_0), \quad \mathbf{Z}_l \sim \mathbb{P}(\mathbf{Z}_l|\mathbf{Z}_{l-1}), \quad l \in \{1, \ldots, L_\mathrm{V} + 1\}, \tag{2}$$

where we denote $\mathbf{Z}_0 := \mathbf{D}$ and $\mathbf{Z}_{L_\mathrm{V}+1} := \mathbf{X}$. While (2) defines the generative pathway, the underlying structure is shaped by the selection principle of (1): the conditional distributions in (2) are implicitly learned if one has learned selection mechanisms in (1) and vice versa.

**Why is this "selection" formulation?** The "selection" perspective is critical for modeling how abstract concepts enforce coherence among their more concrete constituents. Consider generating an image of a "bicycle" (a high-level concept $\mathbf{Z}_l$). Its components – wheels, frame, handlebars (lower-level concepts $\mathbf{Z}_{l+1}$) – must not only be present but also be arranged in a specific, structurally sound configuration. Traditional hierarchical models (Choi et al., 2011; Pearl, 1988; Zhang, 2004) assume independent low-level concepts $Z_{l+1,i}$ given high-level concepts $\mathbf{Z}_l$ and stochastically sample these

components, which could lead to unrealistic arrangements (e.g., wheels detached from the frame if the learned conditional is not perfect). Therefore, these models must additionally incorporate causal edges within each hierarchical level to capture this conditional dependency, resulting in highly dense causal graphs. In contrast, the selection model, by positing that $\mathbf{Z}_l$ is an effect of a specific configuration of $\mathbf{Z}_{l+1}$, emphasizes that the "bicycle" concept arises from a coherent selection and composition of its parts. This structured dependency, induced by the selection mechanism, yields a much simpler graphical model to describe the natural data distribution, thus preferred by the *minimality* principle.

**Connections to text-to-image diffusion models.** The iterative denoising process in diffusion aligns with our hierarchical data construction. These models involve a sequence of transformations $\{f_t\}_{t=1}^{T}$, parameterized by timestep $t$, that progressively restore a less noisy image $\mathbf{X}_t$ from a more corrupted version $\mathbf{X}_{t+1}$. As interpreted by Kong et al. (2024), each $f_{t+1}$ can be viewed as an autoencoder: it extracts a representation $\mathbf{Z}_{\mathcal{S}(t+1)}$ ($\mathcal{S}(t+1)$ indexes U-Net features associated with timestep $t+1$) from the noisy input $\mathbf{X}_{t+1}$, and uses this representation to produce the less noisy $\mathbf{X}_t$. In this view, representations $\mathbf{Z}_{\mathcal{S}(t+1)}$ from higher noise levels (larger $t$, where $\mathbf{X}_{t+1}$ is closer to pure noise) correspond to higher-level, more abstract concepts in our hierarchy (e.g., $\mathbf{Z}_l$ with smaller $l$), as fine-grained details are obscured by noise. Conversely, representations from lower noise levels (smaller $t$) capture more concrete details (e.g., $\mathbf{Z}_l$ with larger $l$). The diffusion model's step-wise refinement thus mirrors our hierarchical generation $\mathbb{P}(\mathbf{Z}_{l+1}|\mathbf{Z}_l)$, with the initial text prompt $\mathbf{D}$ typically guiding the most abstract visual concepts (e.g., $\mathbf{Z}_1 \sim \mathbb{P}(\mathbf{Z}_1|\mathbf{D})$, Figure 1). In our empirical analysis (Section 5, we explicitly map distinct diffusion timesteps to these hierarchical levels: high noise levels (e.g., $t = 899$) and low noise levels (e.g., $t = 100$) to fine-grained details.

**Identifiability and interpretability.** In light of the connection, a crucial question remains: are the internal representations learned by these models (e.g., U-Net features, transformer activations) truly reflective of the ground-truth concepts of the data, or are they merely effective for the generation task without being inherently interpretable and controllable? This motivates the need for *identifiability* guarantees that affirm the equivalence between the two worlds, which we present in Section 4.

## 4 IDENTIFIABLE REPRESENTATIONS UNDER CAUSAL MINIMALITY

We first formally define our core theoretical principle, *causal minimality* (Peters et al., 2017; Spirtes et al., 2000; Hitchcock, 2021): Among all causal models that can explain the observed data, the true model is the simplest one. This principle is the key to our goal of identifiability (Definition 4.1). Causal minimality, as a principle, manifests as concrete, enforceable mechanisms in specific settings. For visual concepts, this mechanism is sparse connectivity in the causal graph (our minimality condition, 4.2-iv), and for text, it is state compression (Condition A.1-ii,iii). Enforcing this sparsity or compression is thus the practical mechanism that provides theoretical guarantees for identifiability.

For visual concepts, minimality manifests as a preference for *sparse graphical dependencies* within the latent hierarchy. This implies that concepts are formed through a limited set of direct causal influences, making the underlying structure easier to discern. In Appendix A, we discuss how the minimality principle translates to seeking the *most compressed representation* for discrete text concepts, the identification theory, and the connection to language models.

A key challenge we address is the identifiability of *hierarchical selection models*. In these models, higher-level concepts are effects of lower-level concepts. This contrasts with traditional hierarchical models where causality often flows from abstract to concrete, and where latent variables typically act as confounders (Pearl, 1988; Zhang, 2004; Choi et al., 2011; Gu & Dunson, 2023; Kong et al., 2024; Xie et al., 2022; Huang et al., 2022; Dong et al., 2023; Kong et al., 2023a; Anandkumar et al., 2013). In our selection framework, latent variables act as colliders, rendering many existing identifiability results inapplicable. This distinction necessitates the novel theoretical development presented herein. Our goal is to achieve *component-wise identifiability*:

**Definition 4.1** (Component-wise Identifiability). Let $\mathbf{Z}$ and $\hat{\mathbf{Z}}$ be variables under two model specifications. We say that $\mathbf{Z}$ and $\hat{\mathbf{Z}}$ are *identified component-wise* if there exists a permutation $\pi$ such that for each $i \in [n(\mathbf{Z})]$, $\hat{Z}_i = h_i(Z_{\pi(i)})$ where $h_i$ is an invertible function.

This strong form of identifiability ensures that each learned latent component $\hat{Z}_i$ corresponds to a single true latent component $Z_{\pi(i)}$. This is vital for unambiguous interpretation and targeted control.

We assume the standard faithfulness condition (Spirtes et al., 2001), meaning the graphical model accurately reflects all conditional independence relations in the data.

In the following, we consider the identification of continuous latent visual concepts $\mathbf{Z}$ and present the counterpart for textual concepts in Appendix A.2.

**Condition 4.2** (Visual Concept Identification Conditions)**.**

> i **Informativeness:** *There exists a diffeomorphism* $g_l : (\mathbf{Z}_l, \boldsymbol{\epsilon}_l) \mapsto \mathbf{X}$ *for* $l \in [0, L]$*, where* $\boldsymbol{\epsilon}_l$ *denotes independent exogenous variables.*

> ii **Smooth Density:** *The probability density function* $p(\mathbf{z}_{l+1}|\mathbf{z}_l)$ *is smooth for any* $l \in [L_V]$*.*

> iii **Sufficient Variability:** *For each* $Z$ *and its parents* $\tilde{\mathbf{Z}} := \mathrm{Pa}(Z)$*, at any value* $\tilde{z}$ *of* $\tilde{\mathbf{Z}}$*, there exist* $n(\tilde{\mathbf{Z}}) + 1$ *distinct values of* $Z$*, denoted as* $\{z^{(n)}\}_{n=0}^{n(\tilde{Z})}$*, such that the vectors* $\mathbf{w}(\tilde{z}, z^{(n)}) - \mathbf{w}(\tilde{z}, z^{(0)})$ *are linearly independent where* $\mathbf{w}(\tilde{z}, z) = \left( \frac{\partial \log p(\tilde{\mathbf{z}}|z)}{\partial \tilde{z}_1}, \ldots, \frac{\partial \log p(\tilde{\mathbf{z}}|z)}{\partial \tilde{z}_{n(\tilde{\mathbf{z}})}} \right)$*.*

> iv **Sparse Connectivity (Minimality):** *For each parent concept* $\tilde{Z}$*, there exists a subset of its children* $\mathbf{Z} \subseteq \mathrm{Ch}(\tilde{Z})$ *such that their* only *common parent is* $\tilde{Z}$*, i.e.,* $\bigcap_{Z \in \mathbf{Z}} \mathrm{Pa}(Z) = \{\tilde{Z}\}$*.*

**Interpreting Condition 4.2.** Condition 4.2-i ensures that the observed data $\mathbf{X}$ (e.g., an image) fully captures the information about the latent concepts $\mathbf{Z}_l$. This is a natural assumption as high-dimensional observations contain rich information. Condition 4.2-ii is a standard regularity assumption for analysis. Both are common in nonlinear ICA literature (Hyvarinen & Morioka, 2016; Hyvarinen et al., 2019; Khemakhem et al., 2020b;a; Von Kügelgen et al., 2021; Kong et al., 2023a). Condition 4.2-iii formalizes the idea that distinct lower-level concepts (e.g., "wheel," "door") respond in sufficiently distinct ways to changes in a shared higher-level concept (e.g., "car"), thus facilitating the identification of these lower-level concepts. Condition 4.2-iv is an instantiation of causal minimality for visual concepts. It posits that the causal graph of concepts is sparse – each concept has a unique "fingerprint" in terms of its connectivities. This sparsity is crucial for disentanglement (Zheng et al., 2022; Lachapelle et al., 2024a; Xu et al., 2024; Lachapelle et al., 2022b;a) and is a less restrictive assumption than, for example, pure observed children for each latent variable (Arora et al., 2012; 2013; Moran et al., 2021). This condition formalizes a core principle: concepts are learned through *comparison*. A concept is identifiable only if the data is rich enough to distinguish it from alternatives. For instance, if "Knight" and "Horse" always co-occur, they are learned as a fused concept; learning them separately requires data that breaks this correlation.

**Theorem 4.3** (Visual Concept Identification)**.** *Assume the process for visual concepts in* (2)*. If a model specification* $\boldsymbol{\theta}_V$ *satisfies Condition 4.2, and an alternative specification* $\hat{\boldsymbol{\theta}}_V$ *satisfies Conditions 4.2-i and 4.2-ii, along with a sparsity constraint such that for corresponding* $\hat{Z}$ *and* $Z$*:*

$$n(\mathrm{Pa}(\hat{Z})) \leq n(\mathrm{Pa}(Z)), \tag{3}$$

*then, if both models* $\boldsymbol{\theta}_V$ *and* $\hat{\boldsymbol{\theta}}_V$ *generate the same observed data distribution* $\mathbb{P}(\mathbf{X})$*, the latent visual concepts* $\mathbf{Z}_l$ *are component-wise identifiable for every level* $l \in [L_V]$*.*

**Proof sketch for Theorem 4.3.** The proof proceeds by identifying the hierarchical model level by level, from the top (most abstract concepts) $\mathbf{Z}_1$ downwards to $\mathbf{Z}_{L_V}$. 1) The paired text data $\mathbf{D}$ acts as an auxiliary variable, providing diverse "influences" on the top-level $\mathbf{Z}_1$. Condition 4.2-iii ensures these interventions have distinguishable effects. Analogous to techniques in nonlinear ICA (Hyvarinen & Morioka, 2016; Hyvarinen et al., 2019; Kong et al., 2022), each component $D$ allows the identification of the subspace of $\mathbf{Z}_1$ variables it influences. 2) With these subspaces identified, one can identify the intersection of these subspaces (Von Kügelgen et al., 2021; Yao et al., 2023; Kong et al., 2023b). Therefore, if the graphical structure is sufficiently sparse, as specified in Condition A.1-iv, one can identify the top-level latent variable $\mathbf{Z}_1$ component-wise. 3) Once $\mathbf{Z}_1$ is identified, its components can serve as the auxiliary variables to identify the next level, $\mathbf{Z}_2$. This process is repeated iteratively down the hierarchy, identifying $\mathbf{Z}_l$ using the already identified $\mathbf{Z}_{l-1}$.

**Implications for text-to-image diffusion models.** Theorem 4.3 underscores that the *sparsity constraint* (3) *is pivotal for identifying true visual concepts.* In practice, this constraint is instantiated through a two-step process: 1) Level-specific concept learning: We train $K$-sparse SAEs on features at the specific timesteps defined in Section 3. This approximates the sparsity condition required

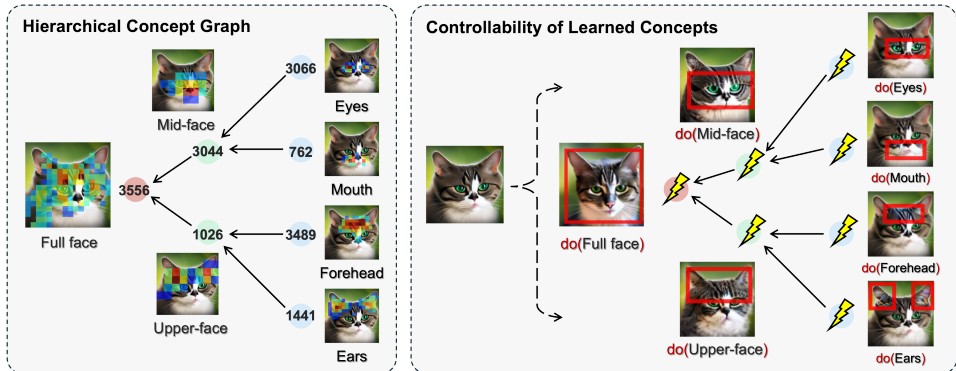

Figure 3: **Examples of hierarchical concept graphs for text-to-image models.** Our method successfully recovers meaningful hierarchical structures, where each node encodes distinct semantic concepts. On the right, we demonstrate feature steering, where manipulating individual nodes leads to changes in the output that align with their position in the hierarchy. Intervening on a high-level concept in the learned graph ("Full face") alters the cat's entire facial structure and fur pattern. In contrast, intervening on a learned lower-level concept (e.g., "Eye") produces a much more localized edit, changing only the shape and color of the eyes while leaving the rest of the face intact. More examples in Appendix F.

by Theorem 4.3. 2) Cross-level causal discovery: We then apply causal discovery algorithms (e.g., PC (Spirtes et al., 2001)) across these sparse features to construct the hierarchical graph, validating that the learned representations align with the theoretical identification guarantees.

## 5 EXPERIMENTS

We present results on T2I models and refer readers to Appendix A.3 for LM experiments.

**Evaluation design and objectives.** We design our experiments to validate our theoretical framework in two ways. In Section 5.1, we provide a direct empirical test of our theory: we apply the sparsity constraints derived from causal minimality (Condition 4.2) and show that we can, as predicted, extract a meaningful and interpretable hierarchical concept graph. In Section 5.2, we demonstrate the utility of these identified concepts. If our concepts are truly component-wise identifiable (Definition 4.1), they should be individually controllable. We test this via a suite of challenging downstream tasks—including model unlearning, controllable image generation, and multi-level editing. For example, we compare against state-of-the-art unlearning methods to rigorously benchmark our concept removal capabilities. Our objective is to show that our theory not only finds interpretable concepts but also provides a practical mechanism for fine-grained, reliable model control. More detailed settings for each experiment are provided in their respective subsections.

**Hierarchical causal analysis.** Our theoretical framework motivates an empirical analysis that differs from standard interpretability approaches. Following the framework established in Sections 3 and 4, we apply our two-step identification process to Stable Diffusion (SD) 1.4 (Rombach et al., 2022a) and Flux.1-Schnell (Labs, 2024) (Appendix F). We analyze feature representations at the previously defined timesteps (899, 500, and 100) to extract and verify the hierarchical concept graph.

**Benefits.** This hierarchical perspective provides two main benefits. First, it enables compositional editing. For a complex object like "a textured tree stump", our analysis can distinguish the "stump" (a mid-level concept) from its "texture" (a low-level one), allowing for independent steering. This is a fine-grained control challenging for non-hierarchical methods that tend to learn entangled features (see Table 6). Second, it allows for targeted intervention. By identifying a concept's level, we can inject a steered feature back into the diffusion process only at its corresponding timestep, which helps in reducing the unwanted artifacts that can arise from applying steering globally across all timesteps (see Figure 5). More details in Appendix E and Figure 9.

### 5.1 INTERPRETABILITY ANALYSIS

**Hierarchical concept graph.** Figure 3 illustrates a hierarchical graph learned through our approach (more in Appendix F). On the left, we display activation maps of different SAE features. Brown

| Method | I2P ↓ | RING-A-BELL ↓ | | | | P4D ↓ | UATK ↓ | COCO | |
|---|---|---|---|---|---|---|---|---|---|
| | | K77 | K38 | K16 | AVG | | | FID ↓ | CLIP ↑ |
| SD 1.4 | 17.8 | 85.26 | 87.37 | 93.68 | 88.10 | 98.70 | 69.70 | 16.71 | **31.3** |
| ESD | 2.87 | 20.00 | 29.47 | 35.79 | 28.42 | 15.49 | 2.87 | 18.18 | 30.2 |
| SA | 2.81 | 63.15 | 56.84 | 56.84 | 58.94 | 12.68 | 2.81 | 25.80 | 29.7 |
| CA | 1.04 | 86.32 | 91.69 | 94.26 | 90.76 | 5.63 | 1.04 | 24.12 | 30.1 |
| MACE | 1.51 | 2.10 | **0.00** | **0.00** | **0.70** | 2.82 | 1.51 | **16.80** | 28.7 |
| UCE | 0.87 | 10.52 | 9.47 | 12.61 | 10.87 | 9.86 | 0.87 | 17.99 | 30.2 |
| RECE | 0.72 | 5.26 | 4.21 | 5.26 | 4.91 | 5.63 | 0.72 | 17.74 | 30.2 |
| SDID | 3.77 | 94.74 | 95.79 | 90.53 | 93.68 | 69.54 | 30.99 | 22.16 | 31.1 |
| SLD-MAX | 1.74 | 23.16 | 32.63 | 42.11 | 32.63 | 9.14 | 2.44 | 28.75 | 28.4 |
| SLD-STRONG | 2.28 | 56.84 | 64.21 | 61.05 | 60.70 | 33.10 | 3.10 | 24.40 | 29.1 |
| SLD-MEDIUM | 3.95 | 92.63 | 88.42 | 91.05 | 90.70 | 24.00 | 1.98 | 21.17 | 29.8 |
| SD1.4-NegPrompt | 0.74 | 17.89 | 40.42 | 34.74 | 31.68 | 10.00 | 1.46 | 18.33 | 30.1 |
| SAFREE | 1.45 | 35.78 | 47.36 | 55.78 | 46.31 | 10.56 | 1.45 | 19.32 | 30.1 |
| TRASCE | 0.45 | 1.05 | 2.10 | 2.10 | 1.75 | 3.97 | **0.70** | 17.41 | 29.9 |
| ConceptSteer | 0.36 | 3.16 | 8.42 | 9.47 | 7.02 | 1.99 | 2.11 | 18.67 | 30.8 |
| **Ours** | **0.25** | **1.05** | **0.00** | 2.11 | 1.05 | **0.66** | 2.11 | 17.02 | **31.3** |

Table 1: **Model unlearning comparisons**. Our method delivers competitive results on unlearning tasks without compromising standard text-to-image generation. See Appendix D for details.

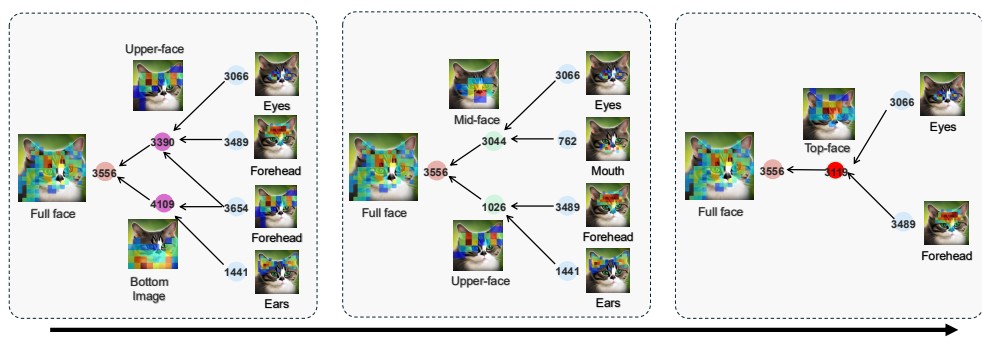

**Sparsity Increasing**

Figure 4: **Ablation studies on the sparsity constraint**. We control feature sparsity at timestep 500. Without enforcing sparsity, the resulting concepts tend to be dense, and the features are less interpretable. Conversely, higher sparsity leads to a more interpretable, sparser graph. However, when sparsity becomes too high, the resulting graph may become overly sparse and fail to adequately capture the generation of the cat face.

nodes (SAE nodes trained on timestep 899) capture high-level features, such as node 3556 representing an entire cat face. Green nodes (timestep 500) reflect mid-level features, like node 3044 capturing the central face. Blue nodes (timestep 100) capture fine details—node 3066 activates on the eyes and node 762 on the mouth. This demonstrates a clear progression from coarse to fine-grained concepts across timesteps. To thoroughly examine the existence of the hierarchical concept graph, we conduct two complementary experiments demonstrating that activations at higher timesteps capture more global semantics, while those at lower timesteps capture more localized details. First, we quantify the spatial spread of activations across timesteps. For each SAE, we compute attribution maps for its top feature indices. Given an SAE feature of shape $64 \times 64 \times 5120$, we compute a $64 \times 64$ attribution map. Applying a 0.1 threshold yields a binary attribution map, from which we measure the proportion of activated pixels. Across 1,000 samples, approximately 280, 630, 880, and 1,400 unique concepts are activated for $K = 1, 3, 5,$ and $10$, respectively. Activations at timestep 899 influence a larger spatial area, indicating that higher timesteps capture more global, distributed concepts. Second, we generate images from 10,000 COCO prompts and deactivate the top-1 SAE activation at each timestep. Comparing the modified generations with the originals shows that deactivations at noisier timesteps cause substantial, global changes, while those at less noisy timesteps produce localized effects. These results confirm that features at different noise levels encode distinct abstraction levels, supporting the hierarchical concept graph.

| (a) Spatial activation spread | | | | (b) SAE-deactivated generation comparison | | | |
|---|---|---|---|---|---|---|---|
| Timestep | Top1 | Top3 | Top5 | Top10 | L1 | LPIPS | CLIP | DINO |
| 100 | 0.27 | 0.21 | 0.19 | 0.15 | 0.004 | 0.002 | 0.999 | 0.999 |
| 500 | 0.30 | 0.25 | 0.21 | 0.17 | 0.013 | 0.020 | 0.995 | 0.993 |
| 899 | 0.53 | 0.41 | 0.33 | 0.24 | 0.070 | 0.220 | 0.948 | 0.903 |

Table 2: **Quantitative analyses across different noise levels.** (a) Spatial activation spread: average proportion of pixels influenced by the top-$k$ SAE activations. Higher timesteps affect a larger spatial area, indicating that SAEs at noisier steps capture more global, distributed concepts. (b) SAE-deactivated generation comparison: similarity metrics between original and SAE-deactivated images. Deactivation at higher timesteps produces greater perceptual and semantic changes, supporting the presence of a hierarchical organization of concepts across timesteps.

| Metric | SD 1.4 | SD1.4 (SAE w/o hier.) | SD1.4 (Ours) |
|---|---|---|---|
| Add tabby pattern – CLIP-I $\downarrow$ | $0.91 \pm 0.05$ | $0.83 \pm 0.07$ | $\mathbf{0.93 \pm 0.04}$ |
| Add tabby pattern – CLIP-T $\uparrow$ | $0.27 \pm 0.00$ | $\mathbf{0.28 \pm 0.02}$ | $\mathbf{0.28 \pm 0.01}$ |
| Add mountains – CLIP-I $\downarrow$ | $0.84 \pm 0.06$ | $0.83 \pm 0.04$ | $\mathbf{0.91 \pm 0.03}$ |
| Add mountains – CLIP-T $\uparrow$ | $\mathbf{0.33 \pm 0.01}$ | $0.32 \pm 0.01$ | $\mathbf{0.33 \pm 0.01}$ |
| Replace rock w/ stump – CLIP-I $\downarrow$ | $0.93 \pm 0.02$ | $0.95 \pm 0.02$ | $\mathbf{0.96 \pm 0.02}$ |
| Replace rock w/ stump – CLIP-T $\uparrow$ | $\mathbf{0.31 \pm 0.01}$ | $0.29 \pm 0.01$ | $\mathbf{0.31 \pm 0.01}$ |

Table 3: **Controllable image generation results**. Our method achieves the best CLIP-I metric, demonstrating greater fidelity to the input images, while reliably executing the target edits.

**Concept steering in hierarchical graphs.** We conduct concept steering using our discovered features, as shown on the right side of Fig. 3 (more in Appendix F). Given a model intermediate feature $x$, the SAE encoder $E$ and decoder $D$ are trained to reconstruct $x$. To steer a specific concept, we obtain the latent representation $z = E(x)$, and extract the steering vector $v$ corresponding to the desired feature. We then modify the original feature to create a steered version $x' = x + \lambda D(v)$, where $\lambda$ modulates the strength. By feeding the steered $x'$ back into the diffusion process at the same timestep, we generate images that reflect the influence of the selected concept. For example, steering node 3556 – associated with the entire face of a cat – results in a significantly altered cat face. Steering the green node 1026 modifies only the upper part of the face, illustrating that it encodes localized information specific to that region.

**Ablation.** As established in the theoretical framework, sparsity is crucial for identifiability. To empirically validate this, we visualize the resulting causal graphs under varying levels of sparsity, as shown in Fig. 4 (more in Appendix F). When sparsity is not enforced, the resulting graph becomes overly dense, making it difficult to interpret and diminishing its semantic clarity. Conversely, imposing excessive sparsity leads to an overly pruned graph that lacks sufficient structure to meaningfully explain the generation process, such as in the case of the cat image. These observations highlight the importance of balancing sparsity to preserve interpretability while maintaining explanatory power.

## 5.2 Downstream Tasks

Thanks to our theoretical framework, we can naturally perform a range of image generation and editing tasks, including model unlearning, controllable image generation, and multi-level editing.

**Model unlearning.** We provide quantitative results of model unlearning on four benchmark datasets: IP2P (Schramowski et al., 2023), three splits of RING-A-BELL (Tsai et al., 2023), P4D (Chin et al., 2023), and UnlearnDiffATK (Zhang et al., 2024b). These benchmarks focus on removing nudity-related concepts, and we report the accuracy of a pretrained nudity detector. Our method achieves the best results across all benchmarks. In addition, to assess whether our method preserves general text-to-image capability, we apply feature steering on normal prompts from MSCOCO (Lin et al., 2014). The 10K results, reflected in low FID and high CLIP scores, demonstrate that our method successfully identifies and removes nudity concepts without affecting unrelated concepts. We also provide results on style removal in the appendix (Table 5) and we achieve superior performance across different metrics and tasks.

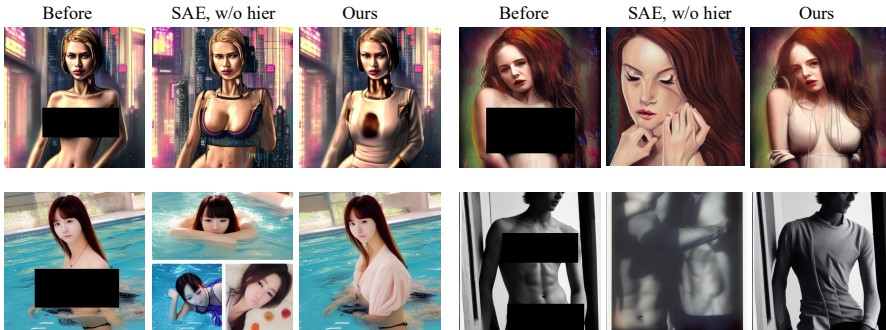

Figure 5: **Generated samples with P4D prompts (Chin et al., 2023).** The Stable Diffusion model is vulnerable to the prompts in the p4d dataset, producing unsafe images. When the hierarchical relationship across timesteps is not considered, negative steering with SAE results in drastic changes to the output. In contrast, our method learns to apply modifications to the nudity feature at a suitable timestep without introducing additional distortions.

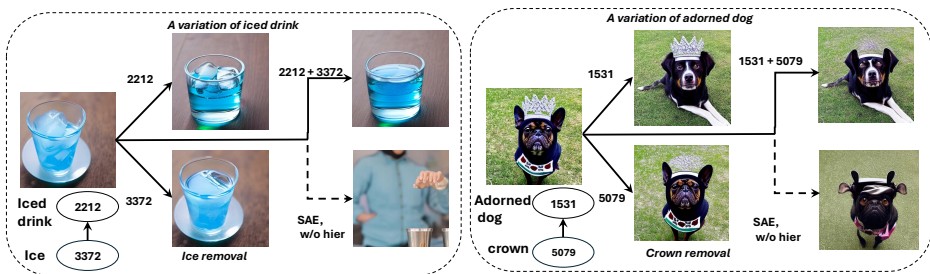

Figure 6: **Examples of multi-level editing** (best viewed with zoom). High-level node 2212 contains all information about the cup, while mid-level node 3372 focuses primarily on the ice cubes. Similarly, high-level node 1531 encompasses all information about the dog (including the crown), and mid-level node 5079 is dedicated to the crown. By modeling hierarchical relationships, we can perform edits that are often difficult to achieve with a single-layer edit. For instance, if we want to generate a variation of the cup while removing the ice cubes, we can apply feature steering on high-level node 2212 to create a new version of the cup, and simultaneously apply negative feature steering on mid-level node 3372 to remove the ice cubes.

**Controllable image generation.** We also evaluate controllable image generation on three editing tasks: adding tabby patterns to cat faces, adding mountains to landscape images, and replacing rocks with textured tree stumps. As shown in Table 3 and Fig.10, our method achieves superior results compared to both the standard text-guided model and SAE without hierarchical modeling.

**Multi-level image editing.** A key advantage of the hierarchical concept graph is that it can combine nodes across different levels for fine-grained image editing. In Fig. 6, to obtain a new drink without ice (while preserving the background), we can apply multi-level editing by steering features at both high-level node 2212 and mid-level node 3372 simultaneously. Without such hierarchical relationship modeling, conventional methods struggle to produce this combination, which can result in undesired changes such as the drink being replaced by a person or the dog's background.

## 6 CONCLUSION

In this work, we present a theoretical framework using causal minimality for identifying latent concepts in hierarchical selection models. We prove that generative model representations can map to true latent variables. Empirically, applying these constraints enables extracting meaningful hierarchical concept graphs from leading models, enhancing interpretability and grounded control. **Limitations:** Our identifiability theorems rely on specific conditions (e.g., faithfulness, smoothness, sparsity) which may not perfectly hold in all real-world models or data; robustness to violations needs further study. Also, scalable causal discovery for high-dimensional concept spaces (e.g., from SAEs) is an important area for future work.

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

# Appendix for "Identifiable Interpretation in Generative Models via Causal Minimality"

**The use of large language models (LLMs).** We employ LLMs to locate typos and polish certain text in the paper. LLMs play no part in the idealization.

## A  FORMULATION, THEORY, AND EXPERIMENTS FOR LANGUAGE MODELS

### A.1  FORMULATION FOR TEXT GENERATION

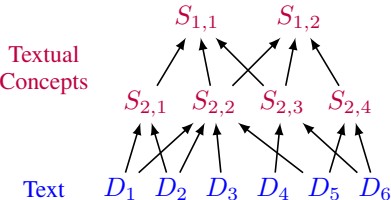

Figure 7: **A textual concept graph.** We denote text as $\mathbf{D}$ and discrete textual concepts as $\mathbf{S}$. High-level concepts function as selection variables of low-level variables.

Textual concepts are $\mathbf{S} := [\mathbf{S}_1, \cdots, \mathbf{S}_{L_T}]$, where $L_T$ is the number of textual hierarchical levels and $\mathbf{S}_l \in \Omega_l \subset \mathbb{N}^{n(\mathbf{S}_l)}$ are concepts at level $l$.

$$\mathbf{S}_1 \sim \mathbb{P}(\mathbf{S}_1), \quad \mathbf{S}_l \sim \mathbb{P}(\mathbf{S}_l|\mathbf{S}_{l-1}), \quad l \in \{2, \ldots, L_T + 1\}, \tag{4}$$

where we denote $\mathbf{Z}_0 := \mathbf{D}$, $\mathbf{Z}_{L_V+1} := \mathbf{X}$, and $\mathbf{S}_{L_T+1} := \mathbf{D}$.

**Connections to autoregressive language models.** An autoregressive language model can be seen as learning an "encoder" that maps a sequence of input tokens ($D_{1:t}$) to an internal state $\hat{\mathbf{S}}_l$ (e.g., activations within transformer layers). This internal state $\hat{\mathbf{S}}_l$ then informs the "decoder" to predict the subsequent token. For optimal prediction, this learned representation $\hat{\mathbf{S}}_l$ should ideally capture the information of the true concept $\mathbf{S}_l$ that *d-separates* the input tokens $D_{1:t}$ from the next token $D_{t+1}$. To achieve this d-separation, $\mathbf{S}_l$ should belong to a higher concept level for a larger span of text $D_{1:t}$ (i.e., larger $t \to$ smaller $l$, see Figure 9). Consequently, broad thematic or narrative structures spanning larger text segments can be compressed into higher-level concepts in our hierarchy (e.g., $\mathbf{S}_1$), while more localized syntactic or lexical choices correspond to lower-level concepts (e.g., $\mathbf{S}_{L_T}$).

**Intuition on "compression" and higher-level concepts in language models.** Our core intuition is that an autoregressive model, at any token position $t$, compresses the *all the preceding sequence* (tokens 1 to $t$) into a representation that is useful for predicting the next token at $t + 1$. In a later position, the model has access to more context and *strictly more* information. Consequently, the minimality constraint promotes more abstract and compressed representations over the information it has seen. This pressure to compress a growing context naturally gives rise to a hierarchy of concepts. Let's use an example for illustration. When a model reads, "He was secretly buying balloons, sending coded messages to friends, and looking up cake recipes...", it would hold onto this list of disparate actions. The meaning is ambiguous; the model has to keep the details in memory. However, once it has parsed the entire sentence, "He was secretly buying balloons, sending coded messages to friends, and looking up cake recipes – he was getting ready for the surprise party for his sister", the model can now form a high-level concept - a celebratory plan — that organizes all the previous, seemingly random actions into a coherent event. This final concept is more compressed and abstract than the initial list of actions, illustrating the move from detailed memorization to a clear, high-level summary as more context becomes available. In this example, the concepts that exist at later stages of the sequence are not just additions but are fundamentally more abstract, as they synthesize a larger body of information. This aligns directly with our theoretical framework (Condition A.1-iii), where we posit that concepts become more compressed (i.e., have minimal support) as we move up the hierarchy.

## A.2 LEARNING TEXTUAL CONCEPTS VIA STATE COMPRESSION

We now turn to the identification of discrete textual concepts $\mathbf{S}$.

The minimality principle manifests as seeking the most "compressed" representation, namely, achieving minimal support sizes for these discrete concepts while preserving full information.

**Condition A.1** (Textual Concept Identification Conditions).

   *i* **Natural Selection**: *Each selection variable $S_l$ has a support $\mathrm{supp}(S_l)$ that is a proper subset of its potential range if its constituent parts (lower-level variables) were combined randomly. That is, $\mathrm{supp}(S_l) \subsetneq f_{\mathbf{D} \to S_l}(\Omega^{n(\mathrm{Pa}(S_l))})$, where $f_{\mathbf{D} \to S_l}$ is the function from $\mathbf{D}$ to $S_l$.*

   *ii* **Bottlenecks**: *The support size of any concept $S_l$ is strictly smaller than the joint support size of its parents $\mathrm{Pa}(S_l)$ in the selection graph.*

   *iii* **Minimal Supports**: *For any $S$, the condition distribution $\mathbb{P}\left(\mathbf{D} \setminus \mathrm{Pa}(S)|S = s, \mathrm{HPa}(S) = \tilde{\mathbf{s}}\right)$ is a one-to-one function w.r.t. the argument $s$.*

   *iv* **No-Twins**: *Distinct latent variables must have distinct sets of adjacent (parent/child) variables.*

   *v* **Maximality**: *The identified latent structure is maximal in the sense that splitting any latent concept variable would violate either the Markov conditions or the No-Twins condition.*

**Interpreting Condition A.1.** Condition A.1-i posits that meaningful text (or textual concepts) occupies a small, structured subset of the vast space of all possible token combinations. We rarely encounter truly random sequences of words in natural language. Conditions A.1-ii and A.1-iii are direct manifestations of causal minimality for discrete concepts. ii implies an information compression moving up the hierarchy—abstract concepts are more succinct. iii demands that each state of a concept $s$ offers unique information about the rest of the text, given its context. Therefore, the representation is most compressed (minimal number of states) and each state contains unique information. Conditions A.1-iv and A.1-v are standard necessary conditions for discrete latent variable model identification (Kivva et al., 2021; 2022), precluding redundant or fragmented latent structures.

**Theorem A.2** (Textual Concept Identification). *Assume the hierarchical process as per* (4). *Let the true underlying parameters be $\boldsymbol{\theta}_{\mathrm{T}}$. If $\boldsymbol{\theta}_{\mathrm{T}}$ satisfies Condition A.1, and an alternative learned model $\hat{\boldsymbol{\theta}}_{\mathrm{T}}$ satisfies Condition A.1-iii, then if both models produce the same observed distribution $\mathbb{P}\left(\mathbf{D}\right)$, the latent textual concepts $\mathbf{S}_l$ are component-wise identifiable for every level $l \in [L_{\mathrm{T}}]$.*

**Proof sketch for Theorem A.2.** The identification for textual concepts proceeds from the bottom level (tokens, $\mathbf{S}_{L_{\mathrm{T}}}$) upwards to the most abstract concepts ($\mathbf{S}_1$). (1) At each level $l + 1$, we make use of the conditional independence relations that the high-level variable $S_{l,i}$ and its hybrid parents $\mathrm{HPa}(S_{l,i})$ d-separate its pure parents $\mathrm{PPa}(S_{l,i})$ from the other variables $\mathbf{S}_l \setminus \{\mathrm{Pa}(S_{l,i})\}$ on level $l$. This relation allows us to identify subsets of $\mathbf{S}_{l+1}$ that share children on level $l$ (Cohen & Rothblum, 1993; Kong et al., 2024) and thus reveals the connectivity between variables in $\mathbf{S}_l$ and $\mathbf{S}_{l+1}$. (2) Once the graphical connections are known, we recover the function $\mathrm{Pa}(S_{l,i}) \mapsto S_{l,i}$ (i.e., how lower-level concepts combine to form $S_{l,i}$). This is done by merging states of $\mathrm{Pa}(S_{l,i})$ that are predictively equivalent. The "Minimal Supports" (Condition A.1-iii) principle dictates that we choose the function that results in the largest equivalence classes over the parent states (i.e., the most compressed representation for $S_{l,i}$). This ensures that the learned concept $\hat{S}_{l,i}$ has the minimum number of necessary states. (3) This process of structure learning and function recovery is repeated from $\mathbf{S}_{L_{\mathrm{T}}}$ (initially using observed tokens $\mathbf{D}$ as $\mathbf{S}_{L_{\mathrm{T}}+1}$) up to $\mathbf{S}_1$, thereby identifying the entire hierarchy.

**Implications for autoregressive language models.** Theorem A.2 suggests that by enforcing a minimality regularization for the most compressed representation (Condition A.1-iii), the learned internal states $\hat{\mathbf{S}}$ of a language model can become equivalent to the underlying textual concepts $\mathbf{S}$. SAEs, when applied to transformer activations, can be seen as a practical way to approximate this minimality. By forcing most latent units to be inactive, SAEs force the model to encode information with the minimal active units, which aligns with our theoretical condition for state compression. This result provides a principled justification for the observed interpretability of SAE-derived features and guides our empirical approach in Section A.3 to extract hierarchical textual concept graphs.

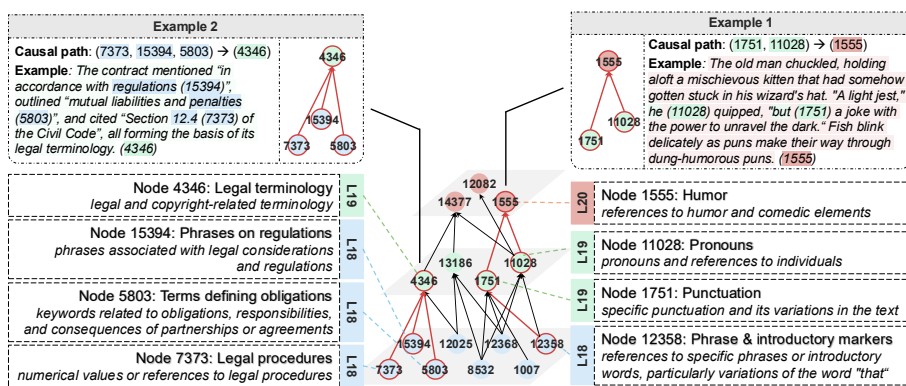

Figure 8: **The learned hierarchical concept graph for autoregressive language models**. By modeling the hierarchy of concepts based on the token sequence order, we recover a meaningful hierarchical graph. The brown nodes (corresponding to later tokens) capture global, high-level information, while the green nodes (from intermediate tokens) represent more localized, lower-level concepts.

## A.3 EXPERIMENTS ON AUTOREGRESSIVE LANGUAGE MODELS

**Implementation.** In this section, we present our implementation for analyzing autoregressive language models. We utilize pretrained SAEs (Bloom et al., 2024) for `Gemma-2-2b-it` (Team, 2024). We partition tokens into three parts based on the their positions in their positions in the input sequence. This segmentation reflects the expectation that tokens convey increasingly abstract or high-level information as the sequence progresses. Finally, we apply causal discovery algorithms to uncover the relationships among features across the different SAEs. More details in Appendix E.

**Results.** Figure 8 shows a learned hierarchical graph (more in Appendix F). Nodes 1555 and 12082 are mostly activated for final tokens in the sequence, and thus capture high-level semantics. Specifically, node 1555 is associated with the humorous tone, while node 12082 represents the role of the dog. Interestingly, node 11028, derived from intermediate tokens, emerges as a causal factor for both 1555 and 12082. This node encodes pronouns and references to individuals, which play a critical role in shaping both the humor and the characterization of the dog.

## B RELATED WORK

**Latent variable identification.** Identifying latent variables is a cornerstone of representation learning. A significant body of work establishes identifiability for single-level latent variable models, often assuming the availability of auxiliary information like domain or class labels (Khemakhem et al., 2020a;b; Hyvarinen & Morioka, 2016; Hyvarinen et al., 2019; Zhang et al., 2024a). Recently, research into language models has explored the linear representation hypothesis, yielding linear-subspace identifiability for latent variables (Reizinger et al., 2024; Liu et al., 2025; Marconato et al., 2024; Rajendran et al., 2024; Jiang et al., 2024). Another research direction (Brady et al., 2023; Lachapelle et al., 2024b;a; Xu et al., 2024; Lachapelle et al., 2022a;b; Zheng et al., 2022; Joshi et al., 2025) leverages sparsity for identification but overlooks the causal relationships among latent variables. Distinct from these approaches, our work formulates the concept space using *hierarchical* models that allow for the explicit modeling of intricate, multi-level conceptual interactions. Our work also connects to the literature on causal abstraction, which studies how a high-level causal model can be faithfully derived from a low-level one (Rubenstein et al., 2017; Geiger et al., 2021; 2024; Beckers & Halpern, 2019; Beckers, 2021). A key distinction is our focus on *component-wise identifiability*, which guarantees that the discovered concepts are equivalent to the true latent variables, providing a stronger foundation for interpretability. Our work is complementary to important research on weak vs. strong (Xi & Bloem-Reddy, 2023) and approximate identifiability (Buchholz & Schölkopf, 2024). While much of this literature analyzes single-level models, our framework is the first to establish component-wise identifiability (Definition 4.1) for hierarchical selection models. This result fits within the weak identifiability category (Xi & Bloem-Reddy, 2023), as do most results in this area. Critically, this level of identifiability is motivated by and sufficient for our downstream tasks, aligning with the principle of "task-identifiability" (Xi & Bloem-Reddy, 2023). It pro-

vides the necessary guarantee for meaningful interpretation and control without requiring the stricter assumptions of strong identifiability. Moreover, the results on approximate identifiability (Buchholz & Schölkopf, 2024) are encouraging, suggesting that robust representations can be learned even if our minimality conditions are only approximately met.

## C  PROOFS

### C.1  PROOF FOR THEOREM 4.3

**Lemma C.1** (Base Case Visual Concept Identification). *Assume the following data-generating process:*

$$\mathbf{C} \sim \mathbb{P}\left(\mathbf{C}|\mathbf{U}\right), \mathbf{V} \sim \mathbb{P}\left(\mathbf{V}\right), \mathbf{X} := g(\mathbf{C}, \mathbf{V}). \tag{5}$$

*We have the following conditions.*

   *i* **Informativeness**: *The function $g(\cdot)$ is a diffeomorphism.*

   *ii* **Smooth Density**: *The probability density function $p(\mathbf{c}, \mathbf{v}|\mathbf{u})$ is smooth.*

   *iii* **Sufficient Variability**: *At any value $\mathbf{c}$ of $\mathbf{C}$, there exist $n(\mathbf{C}) + 1$ distinct values of $\mathbf{U}$, denoted as $\{\mathbf{u}^{(n)}\}_{n=0}^{n(\mathbf{C})}$, such that the vectors $\mathbf{w}(\mathbf{c}, \mathbf{u}^n) - \mathbf{w}(\mathbf{c}, \mathbf{u}^0)$ are linearly independent where $\mathbf{w}(\mathbf{c}, \mathbf{u}) = \left( \frac{\partial \log p(\mathbf{c}|\mathbf{u})}{\partial c_1}, \ldots, \frac{\partial \log p(\mathbf{c}|\mathbf{u})}{\partial c_{n(\mathbf{c})}} . \right)$*

*If a specification $\boldsymbol{\theta}$ satisfies i,ii, and iii, another specification $\hat{\boldsymbol{\theta}}$ satisfies i,ii, and they generate matching distribution $\mathbb{P}\left(\mathbf{X}\right)$, then we can verify that $\mathbf{C}$ and $\hat{\mathbf{C}}$ can be identified up to its subspace.*

*Proof.* Since we have matched distributions, it follows that:

$$p(\mathbf{x}|\mathbf{u}) = \hat{p}(\mathbf{x}|\mathbf{u}). \tag{6}$$

As the generating function $g$ has a smooth inverse (i), we can derive:

$$p(g(\mathbf{c}, \mathbf{v})|\mathbf{u}) = p(\hat{g}(\hat{\mathbf{c}}, \hat{\mathbf{v}})|\mathbf{u}) \implies$$
$$p(\mathbf{c}, \mathbf{v}|\mathbf{u}) \left|\mathbf{J}_{g^{-1}}\right| = \hat{p}(g^{-1} \circ \hat{g}(\hat{\mathbf{c}}, \hat{\mathbf{v}})|\mathbf{u}) \left|\mathbf{J}_{g^{-1}}\right|.$$

Notice that the Jacobian determinant $\left|\mathbf{J}_{g^{-1}}\right| > 0$ because of $g(\cdot)$'s invertibility and let $h := g^{-1} \circ \hat{g} : (\hat{\mathbf{c}}, \hat{\mathbf{v}}) \mapsto (\mathbf{c}, \mathbf{v})$ which is smooth and has a smooth inverse thanks to those properties of $g$ and $\hat{g}$. It follows that

$$p(\mathbf{c}, \mathbf{v}|\mathbf{u}) = \hat{p}(h(\hat{\mathbf{c}}, \hat{\mathbf{v}})|\mathbf{u}) \implies$$
$$p(\mathbf{c}, \mathbf{v}|\mathbf{u}) = \hat{p}(\hat{\mathbf{c}}, \hat{\mathbf{v}}|\mathbf{u}) |\mathbf{J}_{h^{-1}}|.$$

The independence relation in the generating process implies that

$$\log p(\mathbf{c}|\mathbf{u}) + \sum_{i \in [n(\mathbf{v})]} \log p(V_i) = \log \hat{p}(\hat{\mathbf{c}}|\mathbf{u}) + \sum_{i \in [n(\hat{\mathbf{v}})]} \log \hat{p}(\hat{V}_i) + \log |\mathbf{J}_{h^{-1}}|. \tag{7}$$

For any realization $\mathbf{u}^0$, we subtract (7) at any $\mathbf{u} \neq \mathbf{u}^0$ with that at $\mathbf{u}^0$:

$$\log p(\mathbf{c}|\mathbf{u}) - \log p(\mathbf{c}|\mathbf{u}^0) = \log \hat{p}(\hat{\mathbf{c}}|\mathbf{u}) - \log \hat{p}(\hat{\mathbf{c}}|\mathbf{u}^0). \tag{8}$$

Taking derivative w.r.t. $\hat{v}_j$ for $j \in [n(\hat{\mathbf{v}})]$ yields:

$$\sum_{i \in [n(\mathbf{c})]} \frac{\partial}{\partial c_i} (\log p(\mathbf{c}|\mathbf{u}) - \log p(\mathbf{c}|\mathbf{u}^0)) \cdot \frac{\partial c_i}{\partial \hat{v}_j} = 0. \tag{9}$$

The left-hand side zeros out because $\hat{\mathbf{c}}$ is not a function of $\hat{\mathbf{v}}$.

Condition iii ensures the existence of at least $n(\mathbf{c})$ such equations with $\mathbf{u}^1, \ldots, \mathbf{u}^{n(\mathbf{c})}$ that are linearly independent, constituting a full-rank linear system. Since the choice of $j \in [\mathbf{v}]$ is arbitrary. It follows that

$$\frac{\partial c_i}{\partial \hat{v}_j} = 0, \forall i \in [n(\mathbf{c})], j \in [n(\mathbf{v})]. \tag{10}$$

Therefore, the Jacobian matrix $\mathbf{J}_h$ is of the following structure:

$$\mathbf{J}_h = \begin{bmatrix} \frac{\partial \mathbf{v}}{\partial \hat{\mathbf{v}}} & \frac{\partial \mathbf{v}}{\partial \hat{\mathbf{c}}} \\ \frac{\partial \mathbf{c}}{\partial \hat{\mathbf{v}}} & \frac{\partial \mathbf{c}}{\partial \hat{\mathbf{c}}} \end{bmatrix}. \tag{11}$$

(10) suggests that the block $\frac{\partial \mathbf{c}}{\partial \hat{\mathbf{v}}} = 0$. Since $\mathbf{J}_h$ is full-rank, we can deduce that $\frac{\partial \mathbf{c}}{\partial \hat{\mathbf{c}}}$ must have full row-rank and $n(\mathbf{c}) \leq n(\hat{\mathbf{c}})$. The sparsity constraint in (3) further implies that $n(\mathbf{c}) = n(\hat{\mathbf{c}})$. That is, we can correctly identify the dimensionality of the changing subspace $\mathbf{c}$. Moreover, since $\mathbf{J}_h$ is full-rank and the block $\frac{\partial \mathbf{c}}{\partial \hat{\mathbf{v}}}$ is zero, we can derive that the corresponding block $\frac{\partial \hat{\mathbf{c}}}{\partial \mathbf{v}}$ in its inverse matrix $\mathbf{J}_{h^{-1}}$ is also zero. Therefore, there exists an invertible map $\hat{\mathbf{c}} \mapsto \mathbf{c}$, which concludes the proof. $\qquad\square$

**Lemma C.2** (Determining Intersection Cardinality from Union Cardinalities). *Let $\mathcal{A} = \{A_1, A_2, \ldots, A_n\}$ be a finite collection of finite sets. If for any non-empty subset of indices $K \subseteq \{1, 2, \ldots, n\}$, the cardinality of the union $\left|\bigcup_{k \in K} A_k\right|$ is known, then for any non-empty subset of indices $S \subseteq \{1, 2, \ldots, n\}$, the cardinality of the intersection $\left|\bigcap_{s \in S} A_s\right|$ can be determined.*

*Proof.* We proceed by induction on the size of the set of indices $S$, denoted by $|S|$, for which we want to determine the intersection cardinality.

**Base Case:** $|S| = 1$. Let $S = \{i\}$ for some $i \in \{1, 2, \ldots, n\}$. We aim to determine the cardinality $\left|\bigcap_{s \in S} A_s\right| = |A_i|$. The union of a single set $A_i$ is simply $A_i$ itself. That is, $A_i = \bigcup_{k \in \{i\}} A_k$. By the premise of the theorem, the cardinality $\left|\bigcup_{k \in \{i\}} A_k\right|$ is known. Therefore, $|A_i|$ is known. The base case holds.

**Inductive Hypothesis:** Assume that for some integer $m \geq 1$, the cardinality of any intersection of $j$ sets, $\left|\bigcap_{j \in J} A_j\right|$, can be determined from the known union cardinalities for all non-empty index sets $J$ such that $1 \leq |J| \leq m$.

**Inductive Step:** We want to show that the cardinality of any intersection of $m + 1$ sets can be determined. Let $S_{m+1}$ be an arbitrary non-empty subset of indices from $\{1, 2, \ldots, n\}$ such that $|S_{m+1}| = m + 1$. Our goal is to determine $\left|\bigcap_{s \in S_{m+1}} A_s\right|$.

Consider the Principle of Inclusion-Exclusion (PIE) applied to the union of the sets whose indices are in $S_{m+1}$:

$$\left|\bigcup_{s \in S_{m+1}} A_s\right| = \sum_{\emptyset \neq K \subseteq S_{m+1}} (-1)^{|K|-1} \left|\bigcap_{k \in K} A_k\right|$$

This sum runs over all non-empty subsets $K$ of $S_{m+1}$. We can separate the term where $K = S_{m+1}$ (which corresponds to the intersection of all $m + 1$ sets) from the other terms in the sum:

$$\left|\bigcup_{s \in S_{m+1}} A_s\right| = \left(\sum_{\emptyset \neq K \subset S_{m+1}} (-1)^{|K|-1} \left|\bigcap_{k \in K} A_k\right|\right) + (-1)^{|S_{m+1}|-1} \left|\bigcap_{s \in S_{m+1}} A_s\right|$$

Here, the sum is now over all non-empty *proper* subsets $K$ of $S_{m+1}$. We can rearrange this equation to solve for the term $\left|\bigcap_{s \in S_{m+1}} A_s\right|$:

$$(-1)^{|S_{m+1}|-1} \left|\bigcap_{s \in S_{m+1}} A_s\right| = \left|\bigcup_{s \in S_{m+1}} A_s\right| - \sum_{\emptyset \neq K \subset S_{m+1}} (-1)^{|K|-1} \left|\bigcap_{k \in K} A_k\right|$$

Multiplying both sides by $(-1)^{|S_{m+1}|-1}$ (noting that $((-1)^{|S_{m+1}|-1})^2 = 1$):

$$\left|\bigcap_{s \in S_{m+1}} A_s\right| = (-1)^{|S_{m+1}|-1} \left(\left|\bigcup_{s \in S_{m+1}} A_s\right| - \sum_{\emptyset \neq K \subset S_{m+1}} (-1)^{|K|-1} \left|\bigcap_{k \in K} A_k\right|\right)$$

Let us analyze the terms on the right-hand side of this equation:

1. The factor $(-1)^{|S_{m+1}|-1}$ is a known sign, since $|S_{m+1}| = m + 1$.

2. The term $\left|\bigcup_{s \in S_{m+1}} A_s\right|$ is the cardinality of a union of $m + 1$ sets. Since $S_{m+1}$ is a non-empty subset of indices, this value is known by the premise of the theorem.

3. Consider the sum $\sum_{\emptyset \neq K \subset S_{m+1}} (-1)^{|K|-1} \left|\bigcap_{k \in K} A_k\right|$. Each $K$ in this summation is a non-empty proper subset of $S_{m+1}$. Therefore, the size of each such $K$ satisfies $1 \leq |K| \leq m$. By the Inductive Hypothesis, for any such $K$ (i.e., for any intersection of $j$ sets where $1 \leq j \leq m$), the cardinality $\left|\bigcap_{k \in K} A_k\right|$ can be determined from the known union cardinalities. Consequently, every term in this summation, including its sign factor $(-1)^{|K|-1}$, is determinable.

Since all components on the right-hand side of the equation are known or can be determined based on the theorem's premise and the inductive hypothesis, the value of $\left|\bigcap_{s \in S_{m+1}} A_s\right|$ can be determined.

In conclusion, by the principle of mathematical induction, for any non-empty subset of indices $S \subseteq \{1, 2, \ldots, n\}$, the cardinality of the intersection $\left|\bigcap_{s \in S} A_s\right|$ can be determined if the cardinality of any union $\left|\bigcup_{k \in K} A_k\right|$ (for any non-empty $K \subseteq \{1, 2, \ldots, n\}$) is known. $\qquad\square$

**Lemma C.3** (Intersection Block Identification (Kong et al., 2023b)). *We assume the following data-generating process:*

$$[\mathbf{v}_1, \mathbf{v}_2] = g(\mathbf{c}, \mathbf{s}_1, \mathbf{s}_2), \tag{12}$$
$$\mathbf{v}_1 = g_1(\mathbf{c}, \mathbf{s}_1), \tag{13}$$
$$\mathbf{v}_2 = g_2(\mathbf{c}, \mathbf{s}_2), \tag{14}$$

*where $\mathbf{c} \in \mathcal{C} \subset \mathbb{R}^{d_c}$, $\mathbf{s}_1 \in \mathcal{S} \subset \mathbb{R}^{d_{s_1}}$, and $\mathbf{s}_2 \in \mathcal{S}_2 \subset \mathbb{R}^{d_{s_2}}$. Both $g_1$ and $g_2$ are smooth and have non-singular Jacobian matrices almost everywhere, and $g$ is invertible. If $\hat{g}_1 : \mathcal{Z} \to \mathcal{V}_1$ and $\hat{g}_2 : \mathcal{Z} \to \mathcal{V}_2$ assume the generating process of the true model $(g_1, g_2)$ and match the joint distribution $p_{\mathbf{v}_1, \mathbf{v}_2}$, then there is a one-to-one mapping between the estimate $\hat{\mathbf{c}}$ and the ground truth $\mathbf{c}$ over $\mathcal{C} \times \mathcal{S} \times \mathcal{S}$, that is, $\mathbf{c}$ is block-identifiable.*

**Lemma C.4** (One-level Visual Concept Identification). *Assume the process for visual concepts in (2) with $L_V = 1$. If a model specification $\boldsymbol{\theta}_V$ satisfies Condition 4.2, and an alternative specification $\hat{\boldsymbol{\theta}}_V$ satisfies Conditions 4.2-i and 4.2-ii, along with a sparsity constraint such that for corresponding $\hat{Z}$ and $Z$:*

$$n(\mathrm{Pa}(\hat{Z})) \leq n(\mathrm{Pa}(Z)), \tag{15}$$

*then, if both models $\boldsymbol{\theta}_V$ and $\hat{\boldsymbol{\theta}}_V$ generate the same observed data distribution $\mathbb{P}(\mathbf{X})$, the latent visual concepts $\mathbf{Z}_1$ are component-wise identifiable for every level.*

*Proof.* For notational convenience, we denote $\mathbf{Z}_1$ as $\mathbf{S}$ and $\mathbf{D}$ as $\mathbf{U}$ in this proof. This proof consists of two steps. In step one, we identify the connectivity between $U$ and $S$ variables. In step two, we further show the identifiability of the blocks resulting from intersecting the parent sets $\mathrm{Pa}(U)$ of multiple $U$ variables.

**Step 1: connectivity identification.** Since we have access to the joint distribution $\mathbb{P}(\mathbf{S}, \mathbf{U})$, we can derive conditional distributions $\mathbb{P}(\mathbf{S}|\{U_i\}_{i \in \mathcal{H}})$ for any index subset $\mathcal{H} \subseteq [n(\mathbf{U})]$. By Lemma C.1, we can identify the dimensionality of the set of variables $\mathbf{S}$ that are connected to *any* variable in $\{U_i\}_{i \in \mathcal{H}}$ for any $\mathcal{H} \subseteq [n(\mathbf{U})]$. Lemma C.2 implies that we can identify the dimensionality of the set of variables $\mathbf{S}$ that are connected to *all* variables in $\{U_i\}_{i \in \mathcal{H}}$ for any $\mathcal{H} \subseteq [n(\mathbf{U})]$. This information gives rise to a partition of $S$ components, in which each part is connected to the same set of $U$ variables. Therefore, we have identified the bipartite graph between $\mathbf{S}$ and $\mathbf{U}$ up to a permutation.

**Step 2: intersection block identification.** Denote the indices of $S$ variables that are connected to $U_i$ as $\mathcal{I}(i) \subseteq [n(\mathbf{S})]$. We denote the block of $S$ components connected to *all* variables in $\{U_i\}_{i \in \mathcal{H}}$ as $\mathbf{S}_{\cap_{i \in \mathcal{H}} \mathcal{I}(i)}$ for any $\mathcal{H} \subseteq [n(\mathbf{U})]$. Thanks to Lemma C.1, we can identify the block $\mathbf{S}_{\mathcal{I}(i)}$ connected to the variable $U_i$ for any $i \in [n(\mathbf{U})]$. Lemma C.3 allows us to identify the intersection of any two blocks $\mathbf{S}_{\mathcal{I}(i) \cap \mathcal{I}(j)}$ for $i \neq j$. Therefore, repeated applications of Lemma C.3 leads to the identification of the intersection block $\mathbf{S}_{\cap_{i \in \mathcal{H}} \mathcal{I}(i)}$ for any $\mathcal{H} \subseteq [n(\mathbf{U})]$. This concludes the proof. $\qquad\square$

**Condition 4.2** (Visual Concept Identification Conditions).

*i* **Informativeness**: *There exists a diffeomorphism* $g_l : (\mathbf{Z}_l, \boldsymbol{\epsilon}_l) \mapsto \mathbf{X}$ *for* $l \in [0, L]$, *where* $\boldsymbol{\epsilon}_l$ *denotes independent exogenous variables.*

*ii* **Smooth Density**: *The probability density function* $p(\mathbf{z}_{l+1}|\mathbf{z}_l)$ *is smooth for any* $l \in [L_{\mathrm{V}}]$.

*iii* **Sufficient Variability**: *For each* $Z$ *and its parents* $\tilde{\mathbf{Z}} := \mathrm{Pa}(Z)$, *at any value* $\tilde{\mathbf{z}}$ *of* $\tilde{\mathbf{Z}}$, *there exist* $n(\tilde{\mathbf{Z}}) + 1$ *distinct values of* $Z$, *denoted as* $\{z^{(n)}\}_{n=0}^{n(\tilde{Z})}$, *such that the vectors* $\mathbf{w}(\tilde{\mathbf{z}}, z^{(n)}) - \mathbf{w}(\tilde{\mathbf{z}}, z^{(0)})$ *are linearly independent where* $\mathbf{w}(\tilde{\mathbf{z}}, z) = \left( \frac{\partial \log p(\tilde{\mathbf{z}}|z)}{\partial \tilde{z}_1}, \ldots, \frac{\partial \log p(\tilde{\mathbf{z}}|z)}{\partial \tilde{z}_{n(\tilde{\mathbf{z}})}} \right)$.

*iv* **Sparse Connectivity (Minimality)**: *For each parent concept* $\tilde{Z}$, *there exists a subset of its children* $\mathbf{Z} \subseteq \mathrm{Ch}(\tilde{Z})$ *such that their only common parent is* $\tilde{Z}$, *i.e.,* $\bigcap_{Z \in \mathbf{Z}} \mathrm{Pa}(Z) = \{\tilde{Z}\}$.

**Theorem 4.3** (Visual Concept Identification). *Assume the process for visual concepts in* (2). *If a model specification* $\boldsymbol{\theta}_{\mathrm{V}}$ *satisfies Condition 4.2, and an alternative specification* $\hat{\boldsymbol{\theta}}_{\mathrm{V}}$ *satisfies Conditions 4.2-i and 4.2-ii, along with a sparsity constraint such that for corresponding* $\hat{Z}$ *and* $Z$:

$$n(\mathrm{Pa}(\hat{Z})) \leq n(\mathrm{Pa}(Z)), \tag{3}$$

*then, if both models* $\boldsymbol{\theta}_{\mathrm{V}}$ *and* $\hat{\boldsymbol{\theta}}_{\mathrm{V}}$ *generate the same observed data distribution* $\mathbb{P}(\mathbf{X})$, *the latent visual concepts* $\mathbf{Z}_l$ *are component-wise identifiable for every level* $l \in [L_{\mathrm{V}}]$.

*Proof.* By Lemma C.4, we can identify the set of variables $\mathbf{Z}_1$ that are directly connected to the text variables $\mathbf{D}$ and their causal graph. Treating the identified $\mathbf{Z}_1$ as the $\mathbf{U}$ in Lemma C.4, we can further identify $\mathbf{Z}_2$. Repeating this procedure yields the identifiability of the entire model. □

### C.2 PROOF FOR THEOREM A.2

**Definition C.5** (Non-negative Rank). *The non-negative rank of a non-negative matrix* $\mathbf{A} \in \mathbb{R}^{m \times n}$ *is equal to the smallest number* $p$ *such that there exists a non-negative* $m \times p$-*matrix* $\mathbf{B}$ *and a non-negative* $p \times n$-*matrix* $\mathbf{C}$ *such that* $\mathbf{A} = \mathbf{BC}$.

**Lemma C.6** (Conditional Independence and Nonnegative Rank (Cohen & Rothblum, 1993)). *Let* $\mathbf{P} \in \mathbb{R}^{m \times n}$ *be a bi-variate probability matrix. Then its non-negative rank* $\mathrm{rank}_+(\mathbf{P})$ *is the smallest non-negative integer* $p$ *such that* $\mathbf{P}$ *can be expressed as a convex combination of* $p$ *rank-one bi-variate probability matrices.*

**Lemma C.7** (One-level Textual Concept Identification). *Assume the hierarchical process as per* (4) *with* $L_{\mathrm{T}} = 1$. *Let the true underlying parameters be* $\boldsymbol{\theta}_{\mathrm{T}}$. *If* $\boldsymbol{\theta}_{\mathrm{T}}$ *satisfies Condition A.1, and an alternative learned model* $\hat{\boldsymbol{\theta}}_{\mathrm{T}}$ *satisfies Condition A.1-iii, then if both models produce the same observed distribution* $\mathbb{P}(\mathbf{D})$, *the latent textual concepts* $\mathbf{S}_1$ *are component-wise identifiable.*

*Proof.* For each observed variable $D$, we search for the *minimal* set of variables $\mathbf{C} \subseteq (\mathbf{D} \setminus D)$ such that the following conditional independence holds:

$$D \perp \underbrace{\mathbf{D} \setminus (\{D\} \cup \mathbf{C})}_{\mathbf{R}} \,|\, (\mathbf{C}, \mathrm{Ch}(D)). \tag{16}$$

Note that all $D$, $\mathbf{C}$, and $\mathbf{R}$ belong to observed variables, and $\mathrm{Ch}(D)$ is latent. Thanks to Condition A.1-ii and Lemma C.6, we can select $\mathbf{C}$ with which the nonnegative rank of the probability table $\mathbf{T}_{D, \underbrace{\mathbf{D} \setminus (\{D\} \cup \mathbf{C})}_{\mathbf{R}} |\mathbf{C}}$ is strictly smaller than the support size of $D$.

We argue that such $\mathbf{C}$ is the group of variables adjacent to the same variable $S$ at the next level as $D$. In other words, they are the co-parents of $D$, $\mathrm{CoPa}(D)$.

This is because such $\mathbf{C}$ makes 16 hold and thus $\mathrm{CoPa}(D) \subseteq \mathbf{C}$. Otherwise, there would be open paths passing $S$ that induce dependence between $D$ and $\mathrm{CoPa}(D)$, violating the conditional independence relation in (16). Therefore, the minimality constraint would enforce that $\mathbf{C} = \mathrm{CoPa}(D)$. Repeating this procedure to all $D \in \mathbf{D}$, we can construct $\mathbf{S}$ variables at the next level and the adjacency relations between $\mathbf{S}$ and $\mathbf{D}$.

We proceed to identify the function $\mathbf{D} \mapsto \mathbf{S}$. We refer to $D$ as a pure parent if $D$ is adjacent to only one variable $S$ in the discovered graph. For each $S$, we denote its pure parents as $\mathbf{D}^S$ and non-pure parents as $\tilde{\mathbf{D}}^S$. We employ the conditional independence relation $\mathbf{D}^S \perp\!\!\!\perp \mathbf{D} \setminus \mathrm{Pa}(\tilde{\mathbf{D}}^S)|(S, \tilde{\mathbf{D}}^S)$ and Condition A.1-iii to identify the value of $S$, i.e., the function $f_S := (\mathbf{d}^S, \tilde{\mathbf{d}}^S) \mapsto s$.

We first make use of the conditional independence

$$\mathbf{D}^S \perp\!\!\!\perp \mathbf{D} \setminus \mathrm{Pa}(S)|(S, \tilde{\mathbf{D}}^S) \tag{17}$$

to merge the states of pure parents $\mathbf{D}^S$ conditioned on the non-pure parents $\tilde{\mathbf{D}}^S$. Specifically, we condition on non-pure parents $\tilde{\mathbf{D}}^S = \tilde{\mathbf{d}}^S$ for any $\tilde{\mathbf{d}}^S$ present in the support. We define an equivalence relation $\sim$ over values of $(\mathbf{D}^S, \tilde{\mathbf{D}}^S)$ where $(\mathbf{d}_1^S, \tilde{\mathbf{d}}^S) \sim (\mathbf{d}_2^S, \tilde{\mathbf{d}}^S)$ iff they give rise to an identical conditional distribution $\mathbb{P}\left(\mathbf{D} \setminus \mathrm{Pa}(S)|\mathbf{D}^S = \mathbf{d}_1^S, \tilde{\mathbf{D}}^S = \tilde{\mathbf{d}}^S\right) = \mathbb{P}\left(\mathbf{D} \setminus \mathrm{Pa}(S)|\mathbf{D}^S = \mathbf{d}_2^S, \tilde{\mathbf{D}}^S = \tilde{\mathbf{d}}^S\right)$.

We further resort to a more global conditional independence by considering $(\mathbf{D}^S, \tilde{\mathbf{D}}^S)$ as a meta-variable and all the children $\mathrm{Ch}(\tilde{\mathbf{D}}^S)$ associated with this meta-variable:

$$(\mathbf{D}^S, \tilde{\mathbf{D}}^S) \perp\!\!\!\perp \mathbf{D} \setminus \mathrm{Pa}(\mathrm{Ch}(\tilde{\mathbf{D}}^S))|(\mathrm{Ch}(\tilde{\mathbf{D}}^S), \underbrace{\mathrm{Pa}(\mathrm{Ch}(\tilde{\mathbf{D}}^S)) \setminus \{\mathbf{D}^S, \tilde{\mathbf{D}}^S\}}_{:=\tilde{\tilde{\mathbf{D}}}^S}), \tag{18}$$

where $(\mathbf{D}^S, \tilde{\mathbf{D}}^S)$ has become a pure parent of the latent variable $\mathrm{Ch}(\tilde{\mathbf{D}}^S)$. We further group values $([\mathbf{d}^S], \tilde{\mathbf{d}}^S)$ following the rule that $([\mathbf{d}^S]_1, \tilde{\mathbf{d}}_1^S) \sim ([\mathbf{d}^S]_2, \tilde{\mathbf{d}}_2^S)$ iff $\mathbb{P}\left(\mathbf{D} \setminus \mathrm{Pa}(\mathrm{Ch}(\tilde{\mathbf{D}}^S))|([\mathbf{d}^S], \tilde{\mathbf{D}}^S) = ([\mathbf{d}^S]_1, \tilde{\mathbf{d}}_1^S), \tilde{\tilde{\mathbf{D}}}^S = \tilde{\tilde{\mathbf{d}}}^S\right) = \mathbb{P}\left(\mathbf{D} \setminus \mathrm{Pa}(\mathrm{Ch}(\tilde{\mathbf{D}}^S))|([\mathbf{d}^S], \tilde{\mathbf{D}}^S) = ([\mathbf{d}^S]_2, \tilde{\mathbf{d}}_2^S), \tilde{\tilde{\mathbf{D}}}^S = \tilde{\tilde{\mathbf{d}}}^S\right)$ for *each* $\tilde{\tilde{\mathbf{d}}}^S$ on the support. That is, conditioning on any $\tilde{\tilde{\mathbf{d}}}^S$ on the support, $([\mathbf{d}^S]_1, \tilde{\mathbf{d}}_1^S)$ and $([\mathbf{d}^S]_2, \tilde{\mathbf{d}}_2^S)$ cannot be distinguished. Thus, we group them into an equivalence class $[(\mathbf{d}^S, \tilde{\mathbf{d}}^S)]$.

Finally, for each equivalent class $[(\mathbf{d}^S, \tilde{\mathbf{d}}^S)]$, we assign a distinct value $\hat{s}$. This constitutes a function $\hat{f}_S := (\mathbf{d}^S, \tilde{\mathbf{d}}^S) \mapsto \hat{s}$. Due to the deterministic relation from latent variables and their children in (1), $\hat{f}_S$ is well-defined. We denote the random variable $\hat{S} := \hat{f}_S(\mathbf{D}^S, \tilde{\mathbf{D}}^S)$.

In the following, we show that $\hat{S}$ and $S$ are equivalent up to a bijection. We show this by contradiction. Suppose that there existed $(s_0, \hat{s}_0)$ on their respective support, such that their pre-images partially overlapped $(\mathbf{d}_0^S, \tilde{\mathbf{d}}_0^S) \in \hat{f}_S^{-1}(\hat{s}_0) \cap f_S^{-1}(s_0)$ and $\hat{f}_S^{-1}(\hat{s}_0) \neq f_S^{-1}(s_0)$, where $f_S : (\mathbf{d}^S, \tilde{\mathbf{d}}^S) \mapsto s$ represents the true model. Suppose that $f_S^{-1}(s_0)$ missed some elements in $\hat{f}_S^{-1}(\hat{s}_0)$, i.e., $\exists (\mathbf{d}_1^S, \tilde{\mathbf{d}}_1^S) \in \hat{f}_S^{-1}(\hat{s}_0) \setminus f_S^{-1}(s_0)$. In this case, $(\mathbf{d}_0^S, \tilde{\mathbf{d}}_0^S)$ and $(\mathbf{d}_1^S, \tilde{\mathbf{d}}_1^S)$ would lead to distinct values $s_0$ and $s_1$ under model $f_S^{-1}$. By the construction of $\hat{f}_S^{-1}$, this would indicate $\mathbb{P}(\mathbf{D} \setminus \mathrm{Pa}(S)|S = s_0) = \mathbb{P}(\mathbf{D} \setminus \mathrm{Pa}(S)|S = s_1)$ and $\mathbb{P}\left(\mathbf{D} \setminus \mathrm{Pa}(\mathrm{Ch}(\tilde{\mathbf{D}}^S))|S = s_0, \tilde{\tilde{\mathbf{D}}}^S = \tilde{\tilde{\mathbf{d}}}^S\right) = \mathbb{P}\left(\mathbf{D} \setminus \mathrm{Pa}(\mathrm{Ch}(\tilde{\mathbf{D}}^S))|S = s_1, \tilde{\tilde{\mathbf{D}}}^S = \tilde{\tilde{\mathbf{d}}}^S\right)$ for *each* $\tilde{\tilde{\mathbf{d}}}^S$ on the support. Since $s_0 \neq s_1$, this violates Condition A.1-iii, giving rise to a contradiction.

Suppose that $f_S^{-1}(s_0)$ contains additional elements, i.e., $\exists (\mathbf{d}_2^S, \tilde{\mathbf{d}}_2^S) \in f_S^{-1}(s_0) \setminus \hat{f}_S^{-1}(\hat{s}_0)$. In this case, $(\mathbf{d}_0^S, \tilde{\mathbf{d}}_0^S)$ and $(\mathbf{d}_2^S, \tilde{\mathbf{d}}_2^S)$ would lead to one value $s_0$ under model $f_S^{-1}$. By the construction of $\hat{f}_S^{-1}$, this would indicate either $\mathbb{P}\left(\mathbf{D} \setminus \mathrm{Pa}(S)|\mathbf{D}^S = \mathbf{d}_0^S, \tilde{\mathbf{D}}^S = \tilde{\mathbf{d}}_0^S\right) \neq \mathbb{P}\left(\mathbf{D} \setminus \mathrm{Pa}(S)|\mathbf{D}^S = \mathbf{d}_2^S, \tilde{\mathbf{D}}^S = \tilde{\mathbf{d}}_2^S\right)$ or $\mathbb{P}\left(\mathbf{D} \setminus \mathrm{Pa}(\mathrm{Ch}(\tilde{\mathbf{D}}^S))|([\mathbf{d}^S], \tilde{\mathbf{D}}^S) = ([\mathbf{d}^S]_0, \tilde{\mathbf{d}}_0^S), \tilde{\tilde{\mathbf{D}}}^S = \tilde{\tilde{\mathbf{d}}}^S\right) \neq \mathbb{P}\left(\mathbf{D} \setminus \mathrm{Pa}(\mathrm{Ch}(\tilde{\mathbf{D}}^S))|([\mathbf{d}^S], \tilde{\mathbf{D}}^S) = ([\mathbf{d}^S]_2, \tilde{\mathbf{d}}_2^S), \tilde{\tilde{\mathbf{D}}}^S = \tilde{\tilde{\mathbf{d}}}^S\right)$ for some $\tilde{\tilde{\mathbf{d}}}^S$ on the support. By construction of $\hat{f}_S$, this would violate conditional independence (17) or (18) which the graphical structure implies, which leads to a contradiction.

Therefore, we have shown that for each pair $(s, \hat{s})$ on their respective support, their pre-images should be identical as long as they intersect: $\hat{f}_S^{-1}(\hat{s}) \cap f_S^{-1}(s) \neq \emptyset \implies \hat{f}_S^{-1}(\hat{s}) = f_S^{-1}(s)$, which is equivalent to that $\hat{S}$ and $S$ are equivalent up to a bijection. $\qquad\square$

**Condition A.1** (Textual Concept Identification Conditions)**.**

  i **Natural Selection**: *Each selection variable $S_l$ has a support $\operatorname{supp}(S_l)$ that is a proper subset of its potential range if its constituent parts (lower-level variables) were combined randomly. That is, $\operatorname{supp}(S_l) \subsetneq f_{\mathbf{D} \to S_l}(\Omega^{n(\mathrm{Pa}(S_l))})$, where $f_{\mathbf{D} \to S_l}$ is the function from $\mathbf{D}$ to $S_l$.*

  ii **Bottlenecks**: *The support size of any concept $S_l$ is strictly smaller than the joint support size of its parents $\mathrm{Pa}(S_l)$ in the selection graph.*

  iii **Minimal Supports**: *For any $S$, the condition distribution $\mathbb{P}\left(\mathbf{D} \setminus \mathrm{Pa}(S) | S = s, \mathrm{HPa}(S) = \tilde{\mathbf{s}}\right)$ is a one-to-one function w.r.t. the argument $s$.*

  iv **No-Twins**: *Distinct latent variables must have distinct sets of adjacent (parent/child) variables.*

  v **Maximality**: *The identified latent structure is maximal in the sense that splitting any latent concept variable would violate either the Markov conditions or the No-Twins condition.*

**Theorem A.2** (Textual Concept Identification)**.** *Assume the hierarchical process as per* (4)*. Let the true underlying parameters be $\boldsymbol{\theta}_{\mathrm{T}}$. If $\boldsymbol{\theta}_{\mathrm{T}}$ satisfies Condition A.1, and an alternative learned model $\hat{\boldsymbol{\theta}}_{\mathrm{T}}$ satisfies Condition A.1-iii, then if both models produce the same observed distribution $\mathbb{P}\left(\mathbf{D}\right)$, the latent textual concepts $\mathbf{S}_l$ are component-wise identifiable for every level $l \in [L_{\mathrm{T}}]$.*

*Proof.* By Lemma C.7, we can identify the set of variables $\mathbf{S}_1$ adjacent to $\mathbf{D}$ and the bipartite causal graph between these two sets of variables. We then employ the identified $\mathbf{S}_1$ to serve as $\mathbf{D}$ in the first step to identify $\mathbf{S}_2$. Repeating this procedure yields the identifiability of the entire model. $\qquad\square$

# D KEY CONCEPT DISCUSSIONS

**The roles and purposes of "Selection-based hierarchy and causality minimality".** The selection-based hierarchy and causal minimality are constraints on the natural data distribution (images or text), which is a standard modeling practice in causal representation learning (Schölkopf et al., 2021). Specifically, the selection-based hierarchy considers concepts as effects of their constituent parts (Zheng et al., 2024), while causal minimality assumes this underlying causal graph is sparse in a specific way (e.g., Condition 4.2-iv).

**"Innate" hierarchical concept graphs.** "Innate" refers to the causal structure inherent in the natural data-generating process itself. Latent concepts in the real world interact (e.g., 'eyes' and 'nose' are components of a 'face'), forming a pre-existing causal structure which we refer to as the "innate concept graph."

**True latent variables and their verifications.** "True latent variables" follow the standard notion in causal representation learning (Schölkopf et al., 2021): they are the disentangled, interpretable, semantic factors of the real-world data-generating process (e.g., age, object pose). This is in contrast to a deep learning model's learned features, which are often an entangled, uninterpretable mixture optimized for a specific training objective. Aligning learned features with true latent variables (referred to as "identification") is the central goal, as it enables reliable interpretation (e.g., "this feature is age") and precise control (e.g., "increase this feature to make the face older"). This is a fundamental question that our work addresses through both theoretical guarantees and empirical validation. Our work provides the guarantee that if the data-generating process fulfills the property of causal minimality and our learning objective enforces this (e.g., via sparsity), the model's learned features are provably equivalent to the true latent variables. We then validate this empirically via intervention, a standard practice in causal research (Schölkopf et al., 2021). Our experiments (Figure 3 and Figure 8) show that manipulating the theoretically identified features provides semantic control over the generated output, providing evidence that these features are the meaningful causal levers of the generative process.

**Validity of the conditions.** While assumptions on the unobserved data-generating process may not be validated directly, we have reasoned for the plausibility of our conditions by reflecting on natural

properties of real-world data. Beyond standard regularity assumptions like smoothness and variability (Khemakhem et al., 2020a;b; Hyvarinen & Morioka, 2016; Hyvarinen et al., 2019; Zhang et al., 2024a), our key minimality conditions—Sparse Connectivity (Condition 4.2-iv) for vision and Minimal Supports (Condition A.1-ii,iii) for text—are motivated by the observation that concepts typically arise from a sparse set of causes (Lachapelle et al., 2024a; Xu et al., 2024; Lachapelle et al., 2022b; Zheng et al., 2022; Moran et al., 2021) and that language is inherently structured and compressible (Shannon, 1948; Cover, 1999). Perhaps a more convincing validation is the empirical results. Our experiments provide strong indicative support for these assumptions: by actively enforcing sparsity/compression via SAEs, we successfully extract meaningful concept hierarchies in both vision (Figure 3) and text (Figure 8) that are otherwise dense and not easily interpretable. This success provides support for the usefulness of our overall approach and the validity of our assumptions. We acknowledge that these assumptions, like any in this field, may not hold universally. Fortunately, our strong empirical results suggest they seem effective and plausible for the complex, real-world data we study.

**Concept variable interpretation.** Our theory proves the existence of a clean, one-to-one mapping between a learned feature and a true latent variable. This guarantee is what makes a principled interpretation possible in the first place. The subsequent step—assigning a human-understandable description to this now-identified concept—is intrinsically a task that requires human validation. This is a fundamental aspect of all interpretability research (perhaps modern vision-language models have the potential to automate this process).

**Comparison with recent work (Cywiński & Deja, 2025).** On the technique side, Cywiński & Deja (2025) feature an elegant concept location technique by utilizing the score function, which could significantly benefit our algorithm. For example, we could employ SAeUron (Cywiński & Deja, 2025) to confirm whether our features at various timesteps match the concept location it identifies. Our causal learning algorithm explicitly learns the inter-connectivity among concepts across hierarchical levels. Thus, to modify a part of a high-level concept, we could focus our scope on only the variables connected to this specific high-level concept, which lowers the search complexity. In our experiment example, to implement two changes, "replacing the rock with tree stump" and "adding texture to tree stump", SAeUron may need to perform two independent searches across all timesteps and node indices. Our method can help reduce the search space to only the low-level nodes connected to "tree stump". In addition, pinpointing specific diffusion timesteps to intervene on potentially aids in managing undesirable artifacts. Moreover, our explicit concept graph could also give an interpretable, intuitive characterization of the model's knowledge. On the message side, Cywiński & Deja (2025) propose a novel score function to select the timestep and node index for accurate concept unlearning. Our work's focus is to provide concise and informative theoretical conditions to understand concept learning in both vision and language modalities, with potential applications like concept easing or controllable generation. With this work, we hope the theoretical insights will facilitate the development of refined and dedicated methods in the community.

**Comparison with recent work (Kim et al., 2024).** Revelio (Kim et al., 2024) relies on training a classifier on a specific classification dataset. Revelio trains SAEs and a classifier on a specific dataset (e.g., Caltech-101) to evaluate which features and timesteps are most correlated with class labels. Our work, in contrast, does not involve class labels. Our primary contribution is a hierarchical, causal framework designed to interpret the generative process itself. We apply causal discovery algorithms to discover the causal relationships across different levels of concepts without any class labels. We are able to understand how semantic concepts causally relate to one another across different levels of abstraction to form a coherent output (e.g., how "ear" and "mouth" features causally contribute to a "cat face"). Moreover, Kim et al. (2024) do not perform interventions or analyze the compositional structure of generation, which are the central themes of our paper.

## E  IMPLEMENTATION DETAILS

We present the diagram of our method in Fig.9.

**Annotation of the concepts** To annotate the concepts discovered by SAEs, we use a two-step process: 1. Identify concept-related features. For a target concept (e.g., nudity in the unlearning task), we collect a set of prompts related to that concept, generate the corresponding images, and extract the top-K activated feature indices that are consistently triggered across these samples. These shared

indices are treated as related to the target concept. 2. Explore causal relationships. After identifying a node (feature) with first step, we use the inferred causal graph to find its parent and child nodes—features closely related to that concept. We then visualize the node's attribute map (e.g., distinct regions of the cat in Fig.1) to interpret and confirm the concept's semantics.

**Computing resources.** We use one L40 GPU for training the SAEs and a standard MacBook Pro with an M1 chip for causal discovery. Training one SAE takes around 8 hours.

**Vision experiments.** For the diffusion sampling process, we utilize the `sde-dpmsolver++` (Lu et al., 2022) sampler, which adds stochasticity between successive steps. We train the K-sparse autoencoder using a latent dimension of 5120, a batch size of 4096, and the Adam optimizer with a learning rate of 0.0001, setting $K = 10$. We use prompts from the Laion-COCO dataset (Schuhmann et al., 2022).

Our causal discovery procedure consists of the following steps:

1. **Identify key features for each SAE:**
   For every SAE trained at a specific noise level, we first extract the *top-K* feature indices that show the highest average activation on the 10K LAION-COCO subset dataset.

2. **Construct binary feature representations:**
   We then enumerate all unique feature indices across samples. For each sample, we create a binary feature vector where a value of 1 indicates that the corresponding feature index appears in the sample's *top-K* list, and 0 otherwise. This results in a *feature–index matrix* representing the activation pattern of features for each SAE at each noise level.

3. **Apply causal discovery:**
   Using the constructed matrices, we employ the classical causal discovery PC algorithm to infer the causal structure among the feature indices across noise levels. *[PC]* identifies potential directional dependencies, revealing how certain features may causally influence others.

For the sparsity ablation study, we control the top-K value used in the SAE. Specifically, we train additional SAEs with K=4 and K=100 at timestep 500. To evaluate the effect of sparsity (Figure 4), we then perform causal discovery by replacing the SAE features with K=10 with those from the K=4 or K=100 models. Table 1 evaluates the following baselines: SD1.4 (Rombach et al., 2022b), ESD (Gandikota et al., 2023), SA (Heng & Soh, 2023), CA (Kumari et al., 2023), MACE (Lu et al., 2024), UCE (Gandikota et al., 2024), RECE (Gong et al., 2024), SDID (Li et al., 2024), SLD-MAX (Schramowski et al., 2023), SLD-STRONG (Schramowski et al., 2023), SLD-MEDIUM (Schramowski et al., 2023), SD1.4-NegPrompt (Rombach et al., 2022b), SAFREE (Yoon et al., 2024), TRASCE (Jain et al., 2024), and ConceptSteer (Kim & Ghadiyaram, 2025).

**LLM experiments.** We utilize the pretrained SAEs for `gemma-2-2b-it` available from Gemma-Scope (Team, 2024). To collect features, we use the `pile-10k` corpus (Gao et al., 2020). For each sample, we first exclude padding tokens and divide the remaining meaningful tokens into three sequential segments. The first segment is processed through the SAE at layer 18 to obtain feature indices representing lower-level information. The second segment is passed through the SAE at layer 19 to capture intermediate-level features. The final segment is input to the SAE at layer 20 to extract higher-level features. We then apply the PC algorithm for causal discovery using the feature indices from these three representational levels.

## F ADDITIONAL EMPIRICAL RESULTS

**Extension to Flux.1** Our main experiments are conducted on Stable Diffusion V1.4, which adopts a U-Net architecture. To further validate the generality of our approach, we extend it to Flux.1-Schnell, a 12B text-to-image DiT model. Specifically, we extract features at timesteps 0, 1, 2, and 3 (the model performs inference in only four steps, as it is a distilled model) and train SAEs with the following settings: batch size 4096, learning rate 0.0001, latent dimension 12,288, and top-k = 20. Each SAE is trained on the LAION-COCO dataset for 20,000 steps. We use the last double-stream transformer block (out of 18 double-stream and 38 single-stream blocks) as the feature space (3072 dimensions). We then perform causal discovery to identify causal dependencies among features. For evaluation, following the setup used for SD1.4, we test our method on the unlearning

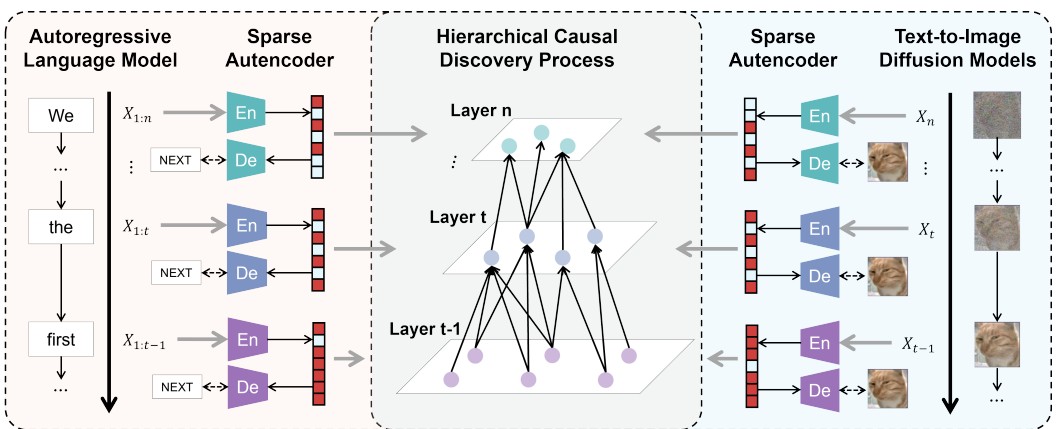

Figure 9: Diagram of our interpretability method. We train SAEs to capture features at different levels (timesteps for diffusion models and token positions for LLMs), and apply causal discovery to construct a hierarchical concept graph.

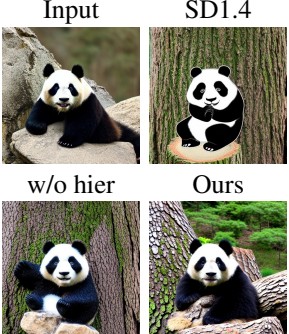

Figure 10: **Examples of controllable image generation.**

| Method | I2P ↓ | RING-A-BELL ↓ | | | | P4D ↓ | UATK ↓ | COCO | |
|--------|-------|------|------|------|------|-------|--------|------|------|
|        |       | K77  | K38  | K16  | AVG  |       |        | FID ↓ | CLIP ↑ |
| **Flux** | 3.08 | 50.53 | 51.58 | 52.63 | 51.58 | 27.15 | 19.72 | 22.89 | 31.57 |
| **Ours-Flux** | 0.94 | 11.58 | 5.26 | 4.21 | 7.01 | 3.31 | 4.93 | 24.40 | 31.54 |

Table 4: **Model unlearning performance on Flux.1-Schnell.** The Flux text-to-image model is susceptible to malicious prompts in benchmark datasets, often producing images containing nudity. By training SAEs on Flux features at different timesteps, we identify the latent representation of the nudity concept and apply negative feature steering to suppress it. This effectively reduces nudity generation while maintaining competitive text-to-image performance on normal prompts from the COCO dataset.

benchmark datasets (Table 4). In particular, we apply negative feature steering at feature index 4390 on timesteps 1, 2, and 3. Our method achieves significantly lower attack success rates on malicious nudity prompts across all benchmarks, demonstrating its robustness and effectiveness on DiT architectures.

Add fire, + Feature 9678

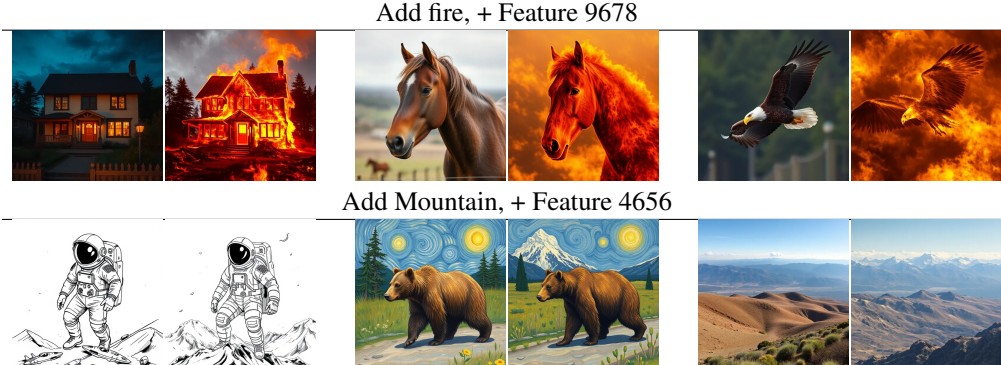

Add Mountain, + Feature 4656

Figure 11: **Context-sensitive and objective-agnostic concepts.** Steering the same concept ("fire" at the top row and "mountain" at the bottom row) yields visual changes in the original context, demonstrating that our learned concepts are *context-sensitive* and *shared across objects/classes*.

**More examples for Figure 3.** Figure 12 and Figure 13 contain more examples of Figure 3. For example, node 3641 in the SAE at timestep 899 contains comprehensive information about the panda, as illustrated by the heatmap. When feature steering is applied, it results in the generation of a new panda. Meanwhile, nodes 1026 and 511 in the SAE at timestep 500 represent different components of the panda. At a finer level of detail, nodes 3489, 3880, and 451 in the SAE at timestep 100 capture specific image features. These hierarchical concept graphs effectively illustrate how the panda is generated.

**More results for model unlearning** In addition to the four benchmark datasets in the main paper, we report results on another commonly used benchmark dataset with two tasks: *Remove Van Gogh* and *Remove Kelly McKernan* in Table.5. We evaluate performance using four metrics: LPIPSe (similarity for prompts with the target style), LPIPSu (similarity for prompts without the style), Acce (how well the target style was removed), and Accu (how well other styles were preserved), with accuracy ratings assessed using GPT-4o. Our method achieves competitive performance across all metrics and tasks.

**Understanding the sparsity constraint.** Figure 14 and Table 6 contain the ablation study for the sparsity constraint. We can observe that a proper sparsity strength can indeed give rise to desirable interpretability results, while too small and too large sparsity constraints may be harmful in practice. As shown in Table 6, a low sparsity penalty results in visualized maps with significant overlap. On the other hand, applying a strong sparsity penalty leads to low node coverage, indicating that the nodes alone are insufficient to fully explain the generation of the entire image.

**More examples for Figure 8.** Figure 15 contains more examples for Figure 8. As discussed in the main paper, we divide the tokens into three segments based on their sequence order, with later

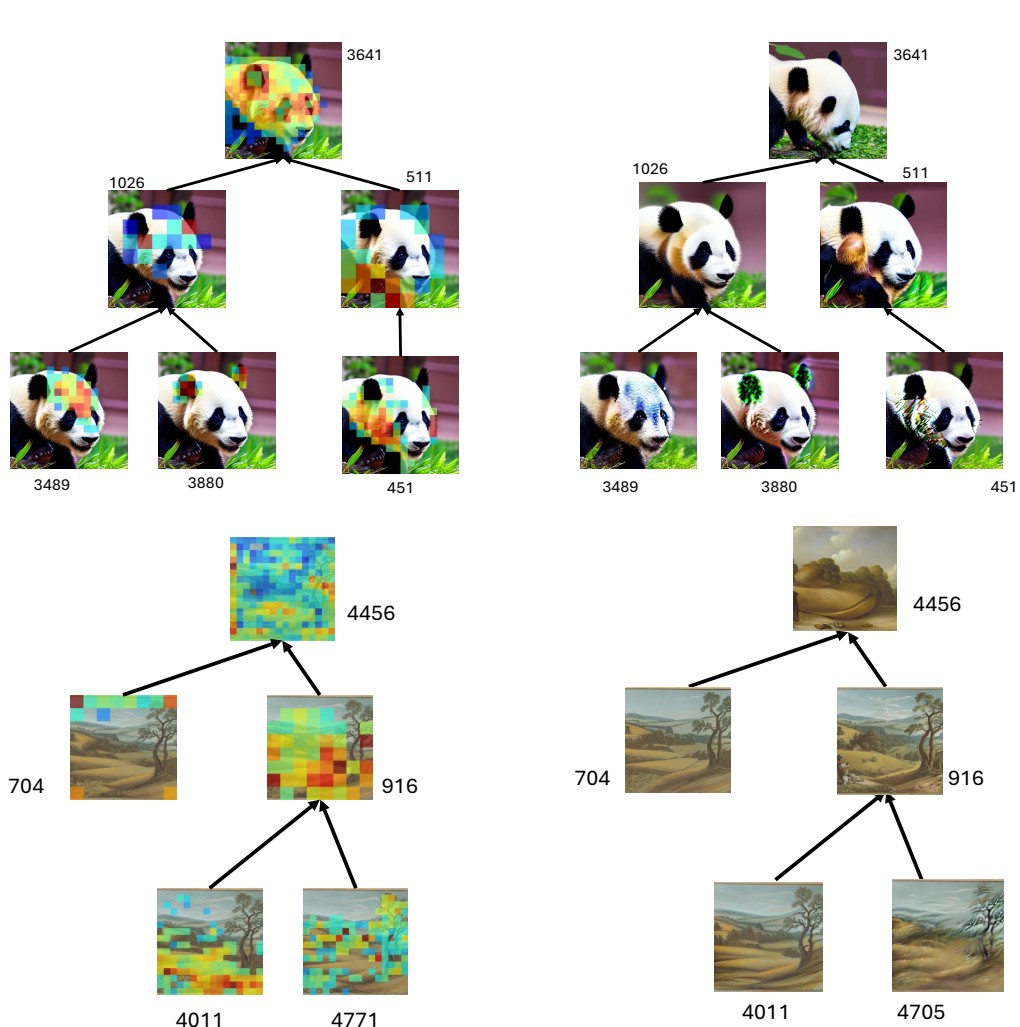

Figure 12: **Discovered hierarchical concept graphs and feature steering visualization for text-to-image generation.** We can observe that features on the hierarchical model represent a part-whole relation, and steering a feature yields corresponding visual variation (e.g., the panda's ears).

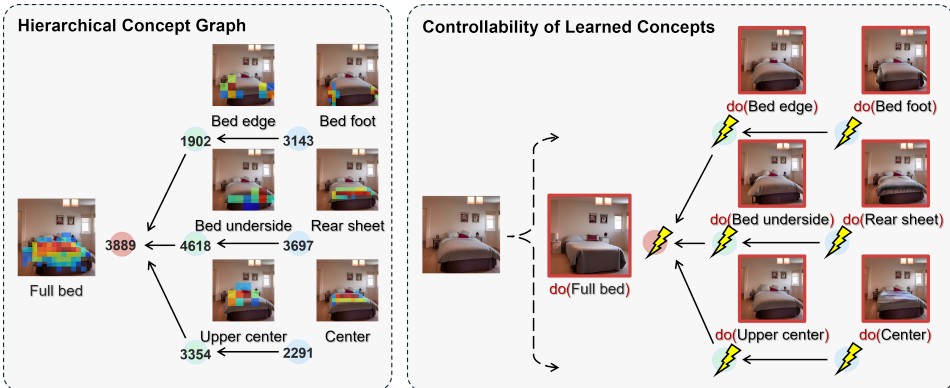

Figure 13: **More examples of the learned hierarchical concept graphs for text-to-image models**. Under appropriate sparsity and noise conditions, our method successfully recovers meaningful hierarchical structures, where each node encodes distinct semantic concepts. On the right, we demonstrate feature steering, where manipulating individual nodes leads to changes in the output that align with their position in the hierarchy – higher-level nodes produce broader semantic shifts, while lower-level nodes control more fine-grained aspects.

| Method | LPIPSe ↑ | LPIPSu ↓ | Acce ↓ | Accu ↑ |
|---|---|---|---|---|
| **Task: Remove "Van Gogh"** | | | | |
| SD-v1.4 | – | – | 0.95 | 0.95 |
| CA (Kumari et al., 2023) | 0.30 | 0.13 | 0.65 | 0.90 |
| RECE (Gong et al., 2024) | 0.31 | 0.08 | 0.80 | 0.93 |
| UCE (Gandikota et al., 2024) | 0.25 | 0.05 | 0.95 | 0.98 |
| SLD-Medium (Schramowski et al., 2023) | 0.21 | 0.10 | 0.95 | 0.91 |
| SAFREE (Yoon et al., 2024) | 0.42 | 0.31 | 0.35 | 0.85 |
| Ours | **0.53** | 0.26 | **0.30** | 0.88 |
| **Task: Remove "Kelly McKernan"** | | | | |
| SD-v1.4 | – | – | 0.80 | 0.83 |
| CA (Kumari et al., 2023) | 0.22 | 0.17 | 0.50 | 0.76 |
| RECE (Gong et al., 2024) | 0.29 | 0.04 | 0.55 | 0.76 |
| UCE (Gandikota et al., 2024) | 0.25 | 0.03 | 0.80 | 0.81 |
| SLD-Medium (Schramowski et al., 2023) | 0.22 | 0.18 | 0.50 | 0.79 |
| SAFREE (Yoon et al., 2024) | 0.40 | 0.39 | 0.40 | 0.78 |
| Ours | **0.48** | 0.20 | **0.35** | 0.81 |

Table 5: **Results on style removal.** We apply negative feature steering to the node to suppress the styles in the image.

tokens expected to encode higher-level information—consistent with the behavior of autoregressive language models. At the highest level, node 11859 represents the "yell mode," characterized by capitalized words conveying a strong tone. The green node 1033, located at an intermediate sequence position, emphasizes importance or intensity—typically a component of the yell mode. At the lowest level, nodes 304, 2009, and 2818 capture various aspects and meanings related to the concept of importance.

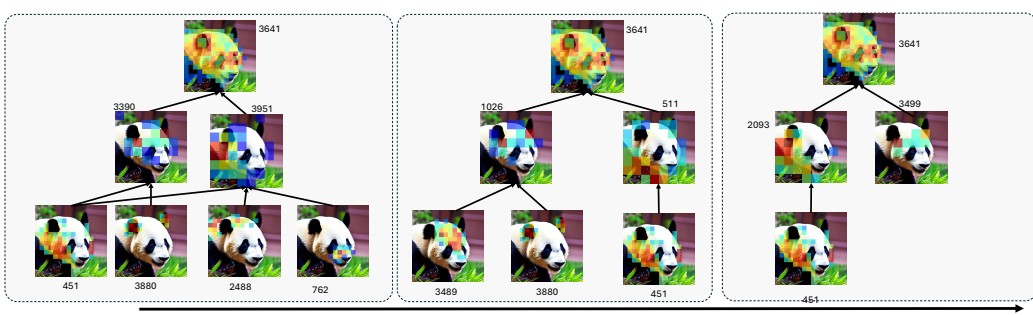

Figure 14: **Understanding the sparsity constraint.** We adjust the top-K value in the SAE at timestep 500 to control the level of sparsity, effectively modifying the sparsity strength of the SAE at this middle layer. As sparsity decreases, the resulting graph becomes denser, introducing many redundant and semantically irrelevant edges. This reduces the overall interpretability of the concept graph. Conversely, increasing sparsity yields a cleaner, more concise graph. However, if sparsity is too high, it may hinder the formation of a complete and interpretable concept graph necessary for image generation.

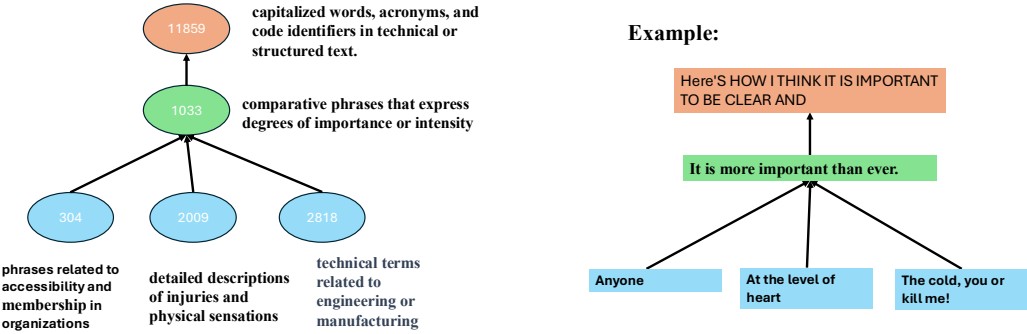

Figure 15: **An example of a discovered hierarchical concept graph for autoregressive language modeling.** Node 11859 represents a "yell mode," characterized by capitalized words that convey a strong tone. The green node 1033 captures the concept of emphasizing importance or intensity. Blue nodes correspond to lower-level information—for instance, node 304 represents entities mentioned throughout the text.

|        | Overlap ↓         | Coverage ↑      |
|--------|-------------------|-----------------|
| K=4    | $0.108 \pm 0.128$ | $26.37 \pm 17.24$ |
| K=10   | $0.089 \pm 0.079$ | $47.90 \pm 12.50$ |
| K=100  | $0.235 \pm 0.132$ | $37.46 \pm 17.31$ |

Table 6: **Quantitative ablation results.** We generate 100 panda images using different random seeds and visualize the feature heatmaps at timestep 500. We adjust the top-K value in the SAE at timestep 500 to control the level of sparsity. To evaluate, we compute the intersection-over-union (IoU) of intermediate heatmaps to measure concept disentanglement, and the union of all features to assess coverage. IoU reflects how distinctly the intermediate concepts are represented, while coverage in percentage indicates the extent to which the intermediate nodes collectively account for the image generation.

