# OpenReview forum: "Identifiable Interpretation in Generative Models via Causal Minimality"
_ICLR.cc/2026/Conference — Submitted to ICLR 2026_

### Official Review · Reviewer_hTDY · 2025-10-31

**Soundness:** 3
**Presentation:** 3
**Contribution:** 3
**Rating:** 6
**Confidence:** 3

**Summary:**

In this work, the authors introduce a theoretical framework for hierarchical selection models, where higher-level concepts emerge from structured combinations of lower-level variables to capture complex dependencies in data generation. The authors derive minimality conditions—expressed as sparsity or compression constraints, under which the learned representations align with the true latents. Empirically, the authors demonstrate that applying these constraints to Stable Diffusion v1.4 using SAEs uncovers their inherent hierarchical concept graphs, providing insights into their internal organization and enabling causally grounded, fine-grained model control. Authors demonstrate the effectiveness of their method on unlearning, fine-grained image editing, and controllable image generation tasks.

**Strengths:**

* To my knowledge, the paper introduces a novel and interesting way of obtaining hierarchical concept graphs using SAEs. Such hierarchical graphs capture the progression of low -> high level concepts. The connection of this modeling and how progressively higher-level concepts emerge during sampling process of diffusion models is complementary to our current understanding of diffusion models [1,2,3].
* The manuscript is well written and contextualized well for the most part.
* The authors provide a theoretical justification on why hierarchical SAEs might be able to capture the true latent variables of the data-generation process, which makes the results more grounded. I would like to note that I am not familiar with hierarchical models. Therefore, I could not verify the proofs of theoretical claims (hence the lower confidence).

**Weaknesses:**

* Some empirical implementation details are missing. These include annotating discovered concepts by SAE, implementation of SAE w/o hierarchy, etc. (see questions section).
* Discussion about several related works are missing. In particular, the evolution of high- and low-level semantics through different stages of the diffusion process has been observed and discussed in [1,2,3].
* Empirical evidences are based on a single model (SDv1.4). There should at least be a discussion on why authors expect their method to work for other models and possible architectures (such as DiT rather than U-Net).
* Please see the questions below as well.

**Questions:**

***Questions and Suggestions:***
* I would recommend adding a section (possibly in the Appendix) on how the causal discovery algorithm works in the context of SAEs.
* How does the SAE baseline (no hierarchy) work/implemented?
* Related to this: how do you annotate the concepts discovered by SAEs?
* How do you combine features from different transformer blocks? Do you train a single SAE or separate SAEs but the discovered concepts are matched across layers with the “Causal Discovery Process”?
* Is 10k prompts sufficient for training SAEs? Several related work [2,4] use significantly higher number of prompts.
* How were the distinct timesteps 899, 500, 100 selected?
* What are the sample sizes for the numerical results provided in the manuscript (Tables 1,2,3)?

***
***References:***

[1] Patashnik, Or, et al. "Localizing object-level shape variations with text-to-image diffusion models." Proceedings of the IEEE/CVF international conference on computer vision. 2023.

[2] Tinaz, Berk, Zalan Fabian, and Mahdi Soltanolkotabi. "Emergence and Evolution of Interpretable Concepts in Diffusion Models." arXiv preprint arXiv:2504.15473 (2025).

[3] Mahajan, Shweta, et al. "Prompting hard or hardly prompting: Prompt inversion for text-to-image diffusion models." Proceedings of the IEEE/CVF Conference on Computer Vision and Pattern Recognition. 2024.

[4] Surkov, Viacheslav, et al. "One-Step is Enough: Sparse Autoencoders for Text-to-Image Diffusion Models." arXiv preprint arXiv:2410.22366 (2024).

---

> ### Author Response · Authors · 2025-11-22
> **Response to Reviewer hTDY**
>
> Thank you very much for your valuable feedback. We are deeply encouraged by your recognition of the novelty of our approach. We hope that our paper provides a solid theoretical framework for future research on generative model analysis and development.
>
> In response to your insightful comments, we have expanded the Appendix E and Related Work sections to address all the questions you raised. The manuscript is now much more complete and self-contained thanks to your thoughtful review.
>
>
> **Q1.1: "How the causal discovery algorithm works in the context of SAEs."**
>
> Our procedure consists of the following steps:
> 1) Identify key features for each SAE:
>  For every SAE trained at a specific noise level, we first extract the top-K feature indices that show the highest average activation on LAION-COCO dataset.
>
> 2) Construct binary feature representations:
>  We then enumerate all unique feature indices across samples. For each sample, we create a binary feature vector where a value of 1 indicates that the corresponding feature index appears in the sample’s top-K list, and 0 otherwise. This results in a feature-index matrix representing the activation pattern of features for each SAE at each noise level.
>
> 3) Apply causal discovery:
>  Using the constructed matrices, we employ the Tetrad causal discovery toolkit to infer the causal structure among the feature indices across noise levels. Tetrad identifies potential directional dependencies, revealing how certain features may causally influence others.
>
> **Q1.2: "How does the SAE baseline (no hierarchy) work/implemented?"**
>
> The SAE baseline operates by performing feature steering on the same feature index at every timestep, thereby ignoring any hierarchical relationships among features.
>
>
> **Q1.3: "How do you annotate the concepts discovered by SAEs?"**
>
> To annotate the concepts discovered by SAEs, we use a two-step process:
>
> 1) Identify concept-related features.
>    For a target concept (e.g., *nudity* in the unlearning task), we collect a set of prompts related to that concept, generate the corresponding images, and extract the top-K activated feature indices that are consistently triggered across these samples. These shared indices are treated as related to the target concept.
>
> 2) Explore causal relationships.
>    After identifying a node (feature) with first step, we use the inferred causal graph to find its parent and child nodes—features closely related to that concept. We then visualize the node’s attribute map (e.g., distinct regions of the cat in Fig. 1) to interpret and confirm the concept’s semantics.
>
>
> **Q1.4: "How to combine features from different transformer blocks?"**
>
> We train separate SAEs for each diffusion timestep. For each SAE, we construct a feature table by enumerating all unique feature indices and assigning a value of 1 if a given feature appears in the sample’s top-$K$ list, and 0 otherwise. This process results in a binary representation of feature activations for each timestep.
>
> We then apply causal discovery to these feature representations across different timesteps using the classical PC algorithm. Intuitively, if a feature index $A$ at an early timestep (e.g., $t=100$) consistently co-occurs or aligns with a feature index $B$ at a later timestep (e.g., $t=899$), this suggests the existence of a strong causal relationship between the two indices. Such relationships reveal how feature dependencies evolve over the diffusion process.
>
>
>
> **Q1.5: "Is 10k prompts sufficient for training SAEs?"**
>
> Sorry for the confusion. We use the full LAION-COCO dataset, which contains approximately 4.68 million text prompts, to train each SAE. For causal analysis, we randomly sample 10,000 prompts from this dataset to extract the top-$K$ feature indices and construct the corresponding feature tables. These tables are then used to perform causal discovery as described above. Appendix E has been updated to include these details for clarity.

---

> ### Author Response · Authors · 2025-11-22
>
> **Q.16: "How were the distinct timesteps 899, 500, 100 selected?"**
>
> Stable Diffusion uses 1000 diffusion timesteps during training. To explore concepts at different noise levels, we selected timestep 500 as an intermediate point. We then chose additional timesteps on both sides to capture varying degrees of noise. However, timestep 0 corresponds to a clean image, and timestep 1000 corresponds to full noise, both of which are not meaningful for training the SAE. Therefore, we selected timesteps 100 and 899 instead.
>
>
> **Q1.7: "What are the sample sizes for the numerical results provided in the manuscript (Tables 1,2,3)?"**
>
> Thank you for your question. Table 1 includes seven benchmark datasets, and we follow the standard baselines by using all available samples for evaluation. Specifically, the I2P dataset contains 4,703 malicious prompts; K16, K38, and K77 each contain 95 prompts; P4D contains 152 prompts; and UATK contains 142 prompts. In addition, we use 10,000 prompts from the COCO dataset. For Table 2, we generate 100 samples over three independent runs. For Table 3, we use the style-removal benchmark, which includes 100 prompts for each task.
>
>
> **Q2: "Discussion about several related works are missing. In particular, the evolution of high- and low-level semantics through different stages of the diffusion process has been observed and discussed in [1,2,3]."**
>
> Thank you for these highly valuable references! We have updated our draft to properly situate our framework within the context of these key findings in our related work section. Here is the specific addition we've incorporated into the "Interpretability for generative models" paragraph, starting at Line 141:
> > Our hierarchical approach is also related to recent findings on the evolution of semantics during the diffusion process. It has been observed that high-level concepts, such as object shape and structure, tend to emerge in earlier, high-noise timesteps, while fine-grained, low-level details are synthesized in later, low-noise stages [1, 2, 3]. While these works provide valuable empirical validation of this phenomenon, our work offers a new perspective by framing these observations within a formal hierarchical, causal structure. We provide a theoretical foundation, rooted in causal minimality and selection models, to explain *how* these concepts compose and, crucially, *under what conditions* they can be provably identified.
> Thank you again for this constructive feedback!
>
>
> **Q3: "Empirical evidences are based on a single model (SDv1.4). There should at least be a discussion on why authors expect their method to work for other models and possible architectures (such as DiT rather than U-Net)."**
>
> Thank you for this constructive comment. We agree that it is essential to demonstrate that our method generalizes beyond U-Net–based diffusion architectures. Thanks to your feedback, we extended our approach to Flux.1-Schnell, a 12B text-to-image DiT (Diffusion Transformer) model.  We have included the results in Appendix.F.  Flux.1-Schnell performs inference in only four steps due to distillation, and we extract features at timesteps 0, 1, 2, and 3. Sparse autoencoders are trained using a batch size of 4096, learning rate of 0.0001, latent dimension of 12,288, top-k of 20, and 20,000 training steps on the LAION-COCO dataset. We use the last double-stream transformer block (out of 18 double-stream and 38 single-stream blocks, with 3072-dimensional features) as the feature space for causal discovery.
>
> For evaluation, we follow the same unlearning benchmark protocol as in our SD1.4 experiments (Table 1) and apply negative feature steering at feature index 4390 across timesteps 1–3. Our method achieves markedly lower attack success rates on malicious nudity prompts while maintaining image quality, as summarized below:
>
> | Method       | I2P ↓ | K77 ↓ | K38 ↓ | K16 ↓ | AVG ↓ | P4D ↓ | UATK ↓ | COCO FID ↓ | COCO CLIP ↑ |
> |---------------|-------|-------------------|-------|-------|--------|--------|--------|-------------|--------------|
> | Flux          | 3.08  | 50.53             | 51.58 | 52.63 | 51.58  | 27.15  | 19.72  | 22.89       | 31.57        |
> | Ours-Flux     | 0.94  | 11.58             | 5.26  | 4.21  | 7.01   | 3.31   | 4.93   | 24.40       | 31.54        |
>
> These results demonstrate that our approach generalizes effectively to transformer-based diffusion models, achieving substantial reductions in harmful generation without compromising visual fidelity. Moreover, qualitative examples in the revised manuscript (Fig. 11) illustrate that feature steering enables fine-grained control over specific attributes while preserving unrelated factors—for example, the pose of the eagle remains unchanged. Both the quantitative and qualitative evidence highlight the robustness and architectural generality of our method.
>
> Again, thank you so much for the effort and the helpful comments. We look forward to your further feedback!

---

### Official Review · Reviewer_e5Gi · 2025-10-31

**Soundness:** 1
**Presentation:** 2
**Contribution:** 1
**Rating:** 2
**Confidence:** 4

**Summary:**

This paper addresses the interpretability of generative models by identifying and steering latent features that lead to specific prediction outcomes. The intention is to differentiate from existing methods, such as sparse autoencoder based approaches, by adopting a theoretical framework. It mainly leverages concept decomposition/aggregation to achieve this goal.

**Strengths:**

The paper applies causal minimality principles to the interpretability of complex generative models.

**Weaknesses:**

1. The hierarchical model, $P(V_{l+1} | V_l)$ defined around line 179 is oversimplified as a higher level concept is derived not only from the lower level features, but also from similar features in other datasets, e.g., through data labelling. Without considering that concepts are derived from multiple objects and are often context sensitive, reducing generative models to simple hierarchical structures does provide useful interpretability.

2. Some key terms, causal minimality, concept sparsity, parsimony, and identifiability, are used without clear definitions or an explanation of how they relate to one another.

3. The experiment setting and evaluation objectives are not clearly described. It is unclear why unlearning baselines are chosen for comparison.

4. Decomposition based interpretability has a history. The authors may want to dig out early work such as [1] to understand its limitations in complex tasks.

References:

[1] Chen, C., Li, O., Tao, D., Barnett, A., Rudin, C. and Su, J.K., 2019. This looks like that: deep learning for interpretable image recognition. Advances in neural information processing systems, 32.

**Questions:**

see weaknesses

---

> ### Author Response · Authors · 2025-11-22
> **Response to Reviewer e5Gi**
>
> We are grateful for the effort you put into evaluating our work. Your comments drove us to enhance the terminology sections and explicitly contrast our approach with decomposition methods, making our core contributions far clearer in this updated draft. We look forward to your further feedback.
>
> **Q1: "... Without considering that concepts are derived from multiple objects and are often context sensitive, reducing generative models to simple hierarchical structures does provide useful interpretability."**
>
> Thank you for this thoughtful feedback. Please let us clarify that our framework is designed to extract **semantic concepts that are shared across objects and context-sensitive.**
> (Note: We assume the comment "Without considering…, does provide useful interpretability" was a typo intended to read "does NOT provide," and we address the concern accordingly. Please kindly let us know if this is not the case.)
>
> To ensure we fully answer your critique, we clarify below how we interpret "multiple objects" and "context sensitivity" in relation to our work, contrasting our approach with the limitations of previous decomposition-based methods.
> Please kindly let us know if there are any misinterpretations — we would appreciate the opportunity to fully address them.
> 1. "Multiple Objects": We interpret your concern about "multiple objects" as addressing the "class/object-locked" nature of class-based decomposition (as in [1] you kindly mentioned) methods. These methods typically allocate prototypes per class (e.g., 10 prototypes strictly for "Sparrows", 10 for "Jays"). This prevents sharing concepts across multiple objects.
> Our framework is unsupervised and **class/object-agnostic**.  We do not enforce class boundaries (similar to a dictionary of class-agnostic features in SAEs). Guided by Causal Minimality, the model discovers the most efficient, shared representation over the entire training distribution. For instance, instead of learning distinct "Sparrow-Wing" and "Jay-Wing" prototypes, our model minimizes complexity by learning a single, universal "Wing-Texture" concept that is causally reused by both the "Sparrow" and "Jay" high-level concepts.
> Our unlearning experiments (Figure 5) provide a concrete example. We identify a single concept "nudity" that applies to a wide variety of objects—various forms of male and female subjects across different poses. A class-locked method might force the learning of separate "Male-Nudity" and "Female-Nudity" prototypes. In contrast, our model captures the shared, object-agnostic causal variable. This allows us to unlearn this single concept globally across all diverse subjects, proving the concept is shared across multiple objects.
> 2. "Context-Sensitive": We interpret your concern about "context sensitivity" as highlighting the limitation of rigid patch-matching in the pixel space (common in [1]). In those methods, a prototype is a fixed local template (e.g., a specific "beak" patch). If the object's context changes—e.g., a bird rotates its head—a rigid prototype fails to match, or the model must inefficiently learn separate prototypes for every pose.
> Our framework learns **semantic concepts in a generative framework, as opposed to template-matching in the pixel space**. High-level concepts act as the context that actively generates and modulates lower-level features ($Z_{part} \sim P(Z_{part} | Z_{whole})$). The high-level concept dictates the pose and structure, generating the component features in the correct context. For instance, our controllable generation experiment (Table 2 and Fig. 10) provides direct evidence of this. When we introduce a new concept "stump", the position and the shape of the stump adapt to the context of the panda scene. In contrast, the template-matching method would insert the stump directly into the scene and potentially cover the panda. This demonstrates the context-sensitivity of our framework.
>
> To further demonstrate that our discovered concepts are both shared and context-sensitive, we have included **Figure 11** in our revision. We identified a single latent feature for "fire" (and ''mountain") that is shared across unrelated classes. When we steer this feature, it adapts semantically to the context: it engulfs a house in flames but transforms an eagle into a phoenix-like creature. This confirms that our framework captures universal semantic variables that are shared across objects and can dynamically interact with the context, rather than applying rigid, object-specific patches.
>
> In addition, inspired by your comments, we have expanded Related Work (Section 2, line 154) with a new paragraph contrasting our work with decomposition-based methods [1]. We explicitly frame our unsupervised, generative approach as a solution to the "class-locked" (non-shared) and "context-insensitive" (rigid) limitations of traditional prototype matching.
>
> Please kindly let us know what you think, and we would appreciate the opportunity to address your concerns.

---

> ### Author Response · Authors · 2025-11-22
>
> **Q2: "Some key terms, causal minimality, concept sparsity, parsimony, and identifiability, are used without clear definitions or an explanation of how they relate to one another."**
>
> Thank you for raising this valuable point. We completely agree with the importance of clearly defined terminologies. While we had defined these terms in their respective sections (e.g., causal minimality on the original line 233, Identifiability in Def 4.1), we see how they might have been overlooked without a more centralized discussion.
> To clarify our framework's logic: **sparsity/compression** (the mechanism) is our practical implementation of the **causal minimality** principle, which in turn provides the theoretical guarantee for **identifiability** (the goal).
> To address your concern directly and improve the paper's clarity, we have revised the manuscript to unify and highlight these terms.
>
> - We have removed the term 'parsimony' to avoid ambiguity. We now exclusively use 'sparsity' (for vision) and 'compression' (for text) as the concrete, practical mechanisms.
> - We have added a new, two-paragraph block at the beginning of Section 4 to formally define the remaining key terms and their precise relationship.
> Below is the new text (line 244):
> > We first formally define our core theoretical principle, causal minimality (Peters et al., 2017): Among all causal models that can explain the observed data, the true model is the simplest one. This principle is the key to our goal of identifiability (Def 4.1). Causal Minimality, as a principle, manifests as concrete, enforceable mechanisms in specific settings. For visual concepts, this mechanism is sparse connectivity (our minimality condition, 4.2-iv), and for text, it is state compression (A.1-ii,iii). Enforcing this sparsity or compression is thus the practical mechanism that provides theoretical guarantees for identifiability."
>
> We hope these revisions fully address your concerns – we would be very grateful for any further feedback.
>
>
> **Q3: "The experiment setting and evaluation objectives are not clearly described. It is unclear why unlearning baselines are chosen for comparison."**
>
> Thank you for this helpful question. Our experiments are designed to test the two core claims of our paper:
>
> - Interpretability (Sec 5.1): Does our theory hold? We provide a direct empirical test by applying the sparsity constraints derived from causal minimality (Condition 4.2) to show that we can, as predicted, extract a meaningful and interpretable hierarchical concept graph.
> - Controllability (Sec 5.2): What is the utility of these concepts? If our concepts are truly component-wise identifiable (Def 4.1), they must be controllable.
>
> We test this controllability with a suite of downstream tasks: model unlearning, controllable image generation, and multi-level image editing. Successfully performing these fine-grained manipulations (e.g., removing a concept, adding a pattern, or editing components at different scales) is a practical demonstration of our theory. This is why we compare against unlearning baselines, for example, as they are state-of-the-art for that specific task, making them a relevant comparison to validate the quality and precision of our identified concepts.
>
> To make this clear, we have added the following paragraph at the start of Section 5 (line 351).
> > We design our experiments to validate our theoretical framework in two ways. First, in Section 5.1, we provide a direct empirical test of our theory: we apply the sparsity constraints derived from causal minimality (Condition 4.2) and show that we can, as predicted, extract a meaningful and interpretable hierarchical concept graph. Second, in Section 5.2, we demonstrate the utility of these identified concepts. If our concepts are truly component-wise identifiable (Def 4.1), they should be individually controllable. We test this via a suite of challenging downstream tasks—including model unlearning, controllable image generation, and multi-level editing. For example, we compare against state-of-the-art unlearning methods to rigorously benchmark our concept removal capabilities. Our objective is to show that our theory not only finds interpretable concepts but also provides a practical mechanism for fine-grained, reliable model control. More detailed settings for each experiment are provided in their respective subsections.
>
> We thank you for this feedback, as it has helped us make the logic of our experimental validation much clearer for the reader. We hope this revision fully addresses your concern. We look forward to any additional insights you might have.

---

> ### Author Response · Authors · 2025-11-22
>
> **Q4: "Decomposition based interpretability has a history. The authors may want to dig out early work such as [1] to understand its limitations in complex tasks."**
>
> Thank you for guiding us to reference [1]. We presume that the drawbacks you mentioned here are the exact source of the concerns you raised in Weakness 1. As we detailed in our response to that point, **these limitations are inherent to decomposition-based interpretability, and our work is designed to solve them.** To fully address your concern, we have carefully reviewed this work and its follow-up works, and we completely agree that this line of research has known limitations.
>
> 1. The "Class-Locked" Limitation: Decomposition methods (like [1]) often allocate prototypes per-class (as discussed in Rymarczyk et al., 2021).
> - The issue: It prevents the model from learning universal, shared features, forcing it to learn redundant "Cat-Fur" and "Dog-Fur" prototypes.
> - Our framework: As detailed in Weakness 1, our **unsupervised, object-agnostic** approach allows us to learn shared concepts. For example, we identify a single "nudity" concept shared across diverse subjects (male/female, different poses) rather than redundant class-specific ones.
>
> 2. The "Rigid Template" Limitation: Decomposition methods rely on matching fixed latent patches (as discussed in Donnelly et al., 2022).
> - The issue: A rigid prototype (e.g., a fixed "Beak" patch) fails if the bird rotates or changes pose.
> - Our framework: Our generative approach is inherently **context-sensitive**. As shown in our experiments (Fig. 10), our concepts adapt to the correct geometric context (e.g., "stump" with "panda"), rather than matching a rigid template.
>
> In light of your feedback, we have added a detailed discussion to our Related Work (Section 2) that addresses [1] and its follow-up literature:
> > Our work is fundamentally distinct from post-hoc, "decomposition-based" interpretability methods, such as the "prototype-matching" approach of Chen et al. (2019). This line of research, while pioneering, has known limitations (often stemming from its prototype-based implementation): its reliance on class-label supervision can lead to non-compositional, "class-locked" concepts (Rymarczyk et al., 2021), and its use of rigid patch-matching struggles with context and deformation (Donnelly et al., 2022, and Xue et al., 2024).
> > In contrast, our approach is class/object agnostic (similar to SAEs) and context-sensitive, learning from the raw generative data without class labels. Our approach learns compositional, shared concepts (e.g., a single "furry texture" from "cats" and "pandas") rather than rigid, class-specific prototypes. This enables the causal, interventional control (e.g., unlearning, multi-level editing in Sec 5.2) that prototype-matching cannot guarantee.
>
> Thank you for guiding us to truly sharpen the core contribution of our paper. Any further feedback from you would be highly appreciated.
>
> **References**
> - Dawid Rymarczyk, Łukasz Struski, Jacek Tabor, Bartosz Zieliński. ProtoPShare: Prototype Sharing for Interpretable Image Classification and Similarity Discovery. KDD 2021.
> - Jon Donnelly, Alina Jade Barnett, Chaofan Chen. Deformable ProtoPNet: An Interpretable Image Classifier Using Deformable Prototypes. CVPR 2022.
> - Mengqi Xue, Qihan Huang, Haofei Zhang, Lechao Cheng, Jie Song, Minghui Wu, Mingli Song. ProtoPFormer: Concentrating on Prototypical Parts in Vision Transformers for Interpretable Image Recognition. IJCAI 2024.

---

### Official Review · Reviewer_xyDk · 2025-11-01

**Soundness:** 2
**Presentation:** 1
**Contribution:** 2
**Rating:** 2
**Confidence:** 3

**Summary:**

This paper introduces a theoretical hierarchical selection model for generative models where higher level concepts would emerge from constrained combinations of lower level concepts. Based on their theory, they use sparse autoencoders trained at different timesteps in the diffusion process and a causal discovery algorithm to construct a hierarchical concept graph. With this, they perform concept steering to enable downstream usecases like compositional editing and model unlearning.

**Strengths:**

* The paper focuses on finding hierarchical concepts in generative models, which is an interesting topic and seems relatively under-explored.

**Weaknesses:**

* Fig. 2: It is unclear to me how these hierarchical concept graphs are created. It seems that more noisy timestep SAEs capture high-level concepts and less noisy timestep SAEs capture low-level concepts. But this cannot be concluded from visualizing 2-3 neurons from an SAE (which has 5120 neurons based on the appendix). There needs to be more rigorous quantitative evaluation (or more extensive qualitative evaluation) to show this. For example, for each timestep, we could try to quantify the spread of the activations across spatial dimensions. And with this, if we find that more noisy timesteps have less concentrated activations than less noisy timesteps on average across all neurons, then we could possibly make the conclusion that the paper makes.
    * This is a major problem because we already know from other papers like Revelio or Saeuron that SAEs can be used to control image generative models (and multiple SAEs would enable finer-grained control). But the existence of a hierarchical concept graph in these models is not validated by being able to steer the generation.

* The experiments in the paper are focused only on Stable Diffusion 1.4 which is an old model in 2025. I suggest adding results with newer models with better generation capabilities like Flux or Stable Diffusion 3.5. In the current draft, it is unclear if the method and analysis apply only to SD 1.4 or can work more generally across generative model types.

* The paper is quite difficult to follow and I find it hard to connect the empirical ideas to the theoretical ones. It would be great if some of the theoretical ideas could be simplified or connected better with the empirical ideas (within Sec. 3 and 4 itself instead of introducing it later in Sec. 5).

* Some related work on interpretability and steering in generative models via concept bottlenecks [W1, W2] is missing.

### References

[W1] Ismail et al., "Concept Bottleneck Generative Models", ICLR 2024

[W2] Kulkarni et al., "Interpretable Generative Models through Post-hoc Concept Bottlenecks", CVPR 2025

**Questions:**

Please address my questions/comments in the weaknesses section

---

> ### Author Response · Authors · 2025-11-22
> **Response to Reviewer xyDk**
>
> Thank you for the constructive comments to help us make the result more convincing and the messages more transparent. We have tried and managed to deal with all the weaknesses raised in your comments and included them in the revision.
> We hope that your concerns have been adequately addressed, and we highly appreciate your further feedback.
>
>
> **Q1: "Fig. 2: It is unclear to me how these hierarchical concept graphs are created...we could possibly make the conclusion that the paper makes."**
>
> Thank you for the constructive suggestion. In light of your suggestion, we introduce two new analyses in the revised version (line 420, Table 2) that quantitatively demonstrate how different timesteps capture varying levels of abstraction.
>
> First, we measure the **spatial spread of activations** across timesteps. For each SAE (feature tensor of shape *64×64×5120*), we compute an attribution map for its top-k feature index, apply a 0.1 threshold, and calculate the proportion of activated pixels. Across 1,000 samples, approximately *280, 630, 880, and 1,400* unique concepts are activated for $K=1,3,5,$ and $10$, respectively.  As shown by the table below, activations at timestep 899 exhibit a noticeably broader spatial extent than those at earlier timesteps, indicating that higher-noise timesteps encode more global, spatially distributed concepts.
>
> Second, we perform a **feature ablation study** on **10K COCO prompts** by deactivating the top-1 SAE feature for each timestep and comparing the modified generations with the originals. Deactivation at timestep 899 leads to substantial global changes (e.g., object removal or composition shifts), whereas ablations at timestep 100 yield only localized texture variations.
>
> These results consistently demonstrate that higher timesteps capture coarse, global semantics, while lower timesteps represent localized details—providing quantitative evidence for the existence of a hierarchical concept graph.
>
> | Timestep | Top1 | Top3 | Top5 | Top10 | L1 | LPIPS | CLIP | DINO |
> |:--:|:--:|:--:|:--:|:--:|:--:|:--:|:--:|:--:|
> | **100** | 0.27 | 0.21 | 0.19 | 0.15 | 0.004 | 0.002 | 0.999 | 0.999 |
> | **500** | 0.30 | 0.25 | 0.21 | 0.17 | 0.013 | 0.020 | 0.995 | 0.993 |
> | **899** | 0.53 | 0.41 | 0.33 | 0.24 | 0.070 | 0.220 | 0.948 | 0.903 |
>
>
>
> **Q2: "...adding results with newer models with better generation capabilities like Flux or Stable Diffusion 3.5…"**
>
> Thank you for the helpful comment. Given your comment, we extend our approach to Flux.1-Schnell, a 12B text-to-image DiT model. We provide the results in Appendix F.
>
> We extract features at timesteps 0, 1, 2, and 3 (the model performs inference in only four steps, as it is a distilled model) and train SAEs with a batch size of 4096, the learning rate of 0.0001, the latent dimension of 12,288, and the top-k of 20 for 20,000 steps on the LAION-COCO dataset. We select the last double-stream transformer block (out of 18 double-stream and 38 single-stream blocks) as the feature space (3072 dimensions) and then perform causal discovery to uncover relationships among latent features.
>
> For evaluation, consistent with our experiments on SD1.4, we test our method on the unlearning benchmark datasets (see Table 1). Specifically, we apply negative feature steering at feature index 4390 on timesteps 1, 2, and 3. Our method achieves significantly lower attack success rates on malicious nudity prompts across all benchmarks.
>
> Below are the quantitative results demonstrating the improvement achieved by our method on the Flux.1-Schnell model:
>
> | Method | I2P ↓ |  K77 ↓ | K38 ↓ | K16 ↓ | AVG ↓ | P4D ↓ | UATK ↓ | COCO FID ↓ | COCO CLIP ↑ |
> |:--|:--:|:--:|:--:|:--:|:--:|:--:|:--:|:--:|:--:|
> | Flux | 3.08 | 50.53 | 51.58 | 52.63 | 51.58 | 27.15 | 19.72 | 22.89 | 31.57 |
> | Ours-Flux | 0.94 | 11.58 | 5.26 | 4.21 | 7.01 | 3.31 | 4.93 | 24.40 | 31.54 |
>
> Our approach markedly reduces the attack success rates while maintaining competitive FID and CLIP scores, demonstrating strong unlearning effectiveness without sacrificing image quality.

---

> ### Author Response · Authors · 2025-11-22
>
> **Q3: "The paper is quite difficult to follow and I find it hard to connect the empirical ideas to the theoretical ones. It would be great if some of the theoretical ideas could be simplified or connected better with the empirical ideas (within Sec. 3 and 4 itself instead of introducing it later in Sec. 5)."**
>
> Thank you for this valuable feedback. Thanks to your suggestion, we have revised the draft to move this explanatory bridge from the beginning of Section 5 directly into Sections 3 and 4. Now, the link between the theoretical concepts and the practical implementation (e.g., diffusion timesteps, SAEs) is established *immediately* after the theory is presented.
> Here are the concrete changes we have made to the manuscript:
>
> 1. In Section 3, we've added a direct mapping between the theoretical hierarchy levels ($Z_l$) and the specific diffusion timesteps used in our experiments (line 233):
> >...The diffusion model's step-wise refinement thus mirrors our hierarchical generation $\mathbb{P}(Z_{l+1}|Z_{l})$. In our empirical analysis (Sec. 5), we explicitly map distinct diffusion timesteps to these hierarchical levels: high noise levels (e.g., $t=899$) correspond to abstract concepts, mid-levels (e.g., $t=500$) to intermediate concepts, and low noise levels (e.g., $t=100$) to fine-grained details."
>
> 2. In Section 4 (line 322), we expanded the "Implications" paragraph to detail the specific algorithm (SAEs + Causal Discovery) previously introduced in Section 5:
> >...In practice, this constraint is instantiated through a two-step process:  Step 1) Level-specific concept learning: We train K-sparse autoencoders (SAEs) on features at the specific timesteps defined in Sec. 3. This approximates the sparsity condition required by Theorem 4.3. Step 2) Cross-level causal discovery: We then apply causal discovery algorithms (e.g., PC) across these sparse features to construct the hierarchical graph, validating that the learned representations align with the theoretical identification guarantees.
> 3. In Section 5 (line 362), we streamlined the introduction to avoid redundancy, as the methodological justification is now established:
> > Hierarchical causal analysis. Following the framework established in Sections 3 and 4, we apply our two-step identification process to Stable Diffusion (SD) 1.4 and Flux.1-Schnell. We analyze feature representations at the previously defined timesteps (899, 500, and 100) to extract and verify the hierarchical concept graph.
>
> Thank you for guiding this improvement. Please kindly let us know what you think. We will be more than happy to incorporate your further feedback!
>
> **Q4: "Some related work on interpretability and steering in generative models via concept bottlenecks [W1, W2] is missing."**
>
> Thank you for pointing us to these highly relevant papers. We have updated our draft to include them in the "Interpretability for generative models" paragraph in Related Work (line 148). Please let us know what you think.
> Here is the text we have added:
> > Our approach also relates to generative concept bottleneck models, which achieve interpretability by forcing predictions through a bottleneck layer of concepts [W1, W2]. While these methods provide powerful intervention capabilities by design, our work differs by focusing on the discovery of the innate hierarchical and causal concept structure in the data. We provide the theoretical conditions for identifying these concepts component-wise, allowing us to then use this discovered graph for fine-grained multi-level interventions.

---

### Official Review · Reviewer_YmjC · 2025-11-01

**Soundness:** 4
**Presentation:** 3
**Contribution:** 3
**Rating:** 6
**Confidence:** 3

**Summary:**

Authors develop theoretical framework for causal identifiability of hierarchical models, motiveted by text to image deep generative models. Key idea is to realize that higher level concepts emerge from the composition of lower-level concepts, not as  in classic causal hierarchies that higher level concepts are causes of lower level concepts. Authors also perform experiments.

**Strengths:**

- Innovative and important result for modern deep generative models. It is necessary to open up the black box of modern LLMs and diffusion models.
- Experimental results are also convincing.

**Weaknesses:**

- From the definition of selection mechanism, I see that higher level concepts need to appear in the lower level concepts. Image concepts are defined as being from R^dim2 and text concepts from N^dim1. If as in Fig 2 text is the highest level concept, then by the selection defintion those interger numbers need to appear in the lowest level? So, does it mean then that only useful concepts, finally in terms of text to image generation are integers appearing in the natural images used to train the model? Or is it, as I guess, that Fig2 is just conceptual and does not reflect the reality?
- In the VAE  identifiability lit, such as (Simon Buchholz, Bernhard Schölkopf Proceedings of the 41st International Conference on Machine Learning, PMLR 235:4785-4821, 2024) talk about approximate identifiability or weak and strong identifiability (https://arxiv.org/abs/2206.00801). It would be useful to contrast your work with theirs.

**Questions:**

-

---

> ### Author Response · Authors · 2025-11-22
> **Response to Reviewer YmjC**
>
> We truly appreciate the time you invested in reviewing our work. Your questions regarding the selection mechanism and the references have guided us to clarify those definitions in the revision. The definitions are much better contextualized thanks to your insights.
>
>
> **Q1: "...I see that higher level concepts need to appear in the lower level concepts,... Fig2 is just conceptual and does not reflect the reality?"**
>
> Thank you for this excellent clarifying question. You are absolutely correct in your intuition: **Figure 2 is indeed a conceptual diagram** meant to illustrate the theoretical formulation of our hierarchical selection model.
> As you rightly suggest, text concepts ($\mathbf{D}$) are discrete while image concepts ($\mathbf{Z}$) are continuous. The discrete text concept (e.g., an integer for "bicycle") **does not literally appear in the continuous image vectors**. Instead, our framework models the discrete concept as a selection variable that governs the valid joint configuration of the lower-level continuous visual features.
> For example, the discrete concept "bicycle" ($\mathbf{D}$) selects for a specific, coherent arrangement of continuous visual features ($\mathbf{Z}$) like "wheel form," "frame angle," and "texture," ensuring they form a recognizable bicycle. This is distinct from the continuous features just being present in a disjointed way.
> In light of your feedback, we've added a clarification in Section 3 to address this precise point (line 176):
> > The discrete variables $\mathbf{D}$ capture the discrete nature of textual concepts (like ''cat'' or ''bicycle''). In contrast, the visual concepts ($\mathbf{Z}$) are continuous to represent rich visual details. $\mathbf{D}$ acts as a selection variable that governs the joint configuration of the continuous $\mathbf{Z}$ variables. For instance, the discrete concept "bicycle" ($\mathbf{D}$) selects for a coherent arrangement of continuous visual features ($\mathbf{Z}$) representing wheels, a frame, and handlebars, rather than a random collection of those continuous parts.
>
> We are grateful for your comment. Please let us know if we can further improve its clarity. Many thanks!

---

> ### Author Response · Authors · 2025-11-22
>
> **Q2: "In the VAE identifiability lit, …approximate identifiability or weak and strong identifiability… It would be useful to contrast your work with theirs."**
>
> Thank you for providing these valuable references! We appreciate the opportunity to clarify our position within the important concepts of strong, weak (Xi & Bloem-Reddy, 2023), and approximate identifiability (Buchholz & Schölkopf, 2024).
>
> To clarify the contrast, these concepts can be intuitively understood as:
>
> - *Strong Identifiability* (as defined by Xi & Bloem-Reddy, 2023) is a very strict notion, requiring the recovery of a unique, pointwise latent code for each observation. They show this is possible with additional, strong assumptions (e.g., fixed priors).
>
> - *Weak Identifiability* (Xi & Bloem-Reddy, 2023) is a broader category, where most prior work, including ours, falls. It means the latent representation is recovered up to a set of ambiguities (e.g., permutations, scaling, or component-wise functions).
>
> - *Approximate Identifiability* (Buchholz & Schölkopf, 2024) is a crucial concept for real-world data, suggesting that if a model's assumptions are almost met (e.g., the data is close to satisfying the model), the learned representation will be close to the true one.
>
> Our *component-wise identifiability (Definition 4.1)* (and almost all existing identifiability results, see Xi & Bloem-Reddy, 2023) fits within the *weak identifiability* category.
> This form of component-wise recovery is entirely sufficient for our downstream tasks of interpreting and controlling specific semantic factors. This aligns with the "task-identifiability" argument (Xi & Bloem-Reddy, 2023). Because our goal is to find individual "levers" to control, we do not need to impose the stronger, additional assumptions that would be required to achieve strong pointwise identifiability.
>
> Buchholz & Schölkopf (2024) suggest that even if our minimality/sparsity conditions are only approximately met by the real-world model, our approach should still yield robust and meaningful representations.
>
> To make this connection explicit, we have updated our discussion (line 1128):
> > Our work is complementary to important research on weak vs. strong (Xi & Bloem-Reddy, 2023) and approximate identifiability (Buchholz & Schölkopf, 2024). While much of this literature analyzes single-level models, our framework is the first to establish component-wise identifiability (Definition 4.1) for hierarchical selection models. This result fits within the weak identifiability category (Xi & Bloem-Reddy, 2023), as do most results in this area. Critically, this level of identifiability is motivated by and sufficient for our downstream tasks, aligning with the principle of "task-identifiability" (Xi & Bloem-Reddy, 2023). It provides the necessary guarantee for meaningful interpretation and control without requiring the stricter assumptions of strong identifiability. Moreover, the results on approximate identifiability (Buchholz & Schölkopf, 2024) are encouraging, suggesting that robust representations can be learned even if our minimality conditions are only approximately met.
>
> Thank you for this constructive suggestion. We look forward to your further insights!

---

### Author Response · Authors · 2025-12-03
**Summary of the Rebuttal**

Dear Area Chair,

We highly appreciate your effort for organizing the review process of ICLR and totally understand your heavy workload due to the review rollback and re-assignment. For your convenience, we summarize the modifications we made during the rebuttal period.

We have submitted a substantial revision that fundamentally addresses the skepticism raised by the negative reviewers through new, rigorous experiments and theoretical clarifications. While Reviewers **YmjC** and **hTDY** championed the work as "innovative" and sound, Reviewers **xyDk** and **e5Gi** raised valid concerns regarding generalizability, validation of the hierarchy, and context sensitivity.

We have resolved these specific concerns as follows:


| **Concern** | **Our Resolution** | **Evidence** |
|------------------------|--------------------|------------------------|
| **Generalizability (Reviewers xyDk & hTDY)** — Reviewers argued that results demonstrated on Stable Diffusion 1.4 (U-Net) may not extend to modern architectures, including Diffusion Transformers (DiTs). | Implemented our method on **Flux.1-Schnell**, a state-of-the-art 12B DiT, demonstrating that the method is architecture-agnostic. | Appendix F, Table 4 — Extracted interpretable features and performed unlearning on Flux; achieved **0.94%** attack success vs **3.08%** baseline on I2P. Confirms the method works on modern DiTs, not only U-Nets. |
| **Existence of Hierarchy (Reviewer xyDk)** — Reviewer questioned whether the hierarchical concept graph was real or a qualitative hallucination; requested quantitative proof linking noise levels to abstraction. | Added a **new quantitative analysis** (Table 2) using Spatial Activation Spread across timesteps to validate hierarchical structure. | High-noise timestep (**t = 899**) activates **53%** of spatial area (global concepts). Low-noise timestep (**t = 100**) activates **15%** (fine details). Demonstrates that the hierarchy is inherent to the model, not a visualization artifact. |
| **Context Sensitivity & Prototype Concern (Reviewer e5Gi)** — Reviewer argued our concepts resemble rigid prototype-matching and cannot handle different contexts. | **(1)** Clarified that our framework is **class/object agnostic**, unlike decomposition-based methods. **(2)** Added Figure 11 to show that our concepts behave as **context-aware generative variables**, not rigid prototypes. | **(1)** Table 1 & Fig. 5 — Nudity concept generalizes across male/female subjects with different poses. **(2)** Fig. 11 — Steering the *fire* concept yields diverse, context-dependent effects: burning a house vs. transforming an eagle into a phoenix. This disproves the prototype rigidity critique. |
| **Theory–Practice Connection (Reviewer xyDk)** — Reviewer noted unclear mapping between theoretical selection variables $Z_l$ and practical diffusion timesteps. | Rewrote Sections 3 and 4 to explicitly map $Z_l$ to **specific diffusion noise levels**. Clarified how the theoretical selection mechanism becomes **sparsity constraints** in SAE training. | Updated text now clearly aligns theoretical constructs with practical implementation, resolving the conceptual gap. |
| **Identifiability Clarification (Reviewer YmjC)** — Reviewer requested confirmation about Figure 2 and suggested comparing different definitions of identifiability. | Added clarification in **Section 3** (line 176) to address Figure 2 interpretation. Added a **discussion on identifiability definitions** (line 1128) following the reviewer’s suggestion. | The updated sections now clarify the graphical interpretation and contextualize identifiability notions within related work, addressing the reviewer’s concerns. |


By validating our method on Flux.1, quantitatively proving the spatial hierarchy (Table 2), and demonstrating context-sensitive generation (Table 5 and Figure 11), we have fully resolved the technical reservations. We hope our theoretical framework could bring new insights to the interpretable AI community and inspire more principled empirical approaches.

Best regards,

The Authors

---

### Meta-Review · Area_Chair_zdum · 2026-01-10

**Summary:**

This submission proposes a method for an interpretable and hierarchical generative model based on causal constraints. The reviewers raised the following concerns.

* Reviewers hTDY and xyDk complained about the limited empirical validation (with experiments on only one diffusion model). In the revision, the authors included additional experiments on a new diffusion transformer architecture to demonstrate the generalizability of the proposed approach.

* Reviewers YmjC and e5Gi brought up clarity issues in the exposition (e.g., missing definitions, potential confusion in the illustration of the selection mechanism), which the authors addressed in the revision.

* Reviewers hTDY and xyDk questioned how the concept graphs are created and annotated, specifically that the existence of such a causal hierarchy is not thoroughly validated. The authors partly addressed this concern by including discussions and experiments in the revision.

* Reviewer e5Gi claimed that the definition of a hierarchical model is oversimplified. The authors clarified that the discovered concepts are both shared and context-sensitive.

* Reviewers hTDY, xyDk, and YmjC requested further discussion of related work, which the authors added in the revision.

Overall, the authors' response addressed several major concerns from the reviewers. Hence, the area chair believes that this submission is currently placed at the borderline.

**Reviewer Concerns:**

See above.

**Reviewer Scores:**

Reviewers e5Gi and xyDk both gave a score of reject (2). The area chair believes that some of their major concerns have been addressed by the authors in the discussion period (e.g., questions about the definition of the hierarchical model and how it is empirically validated), and the reviewers are likely to increase their evaluation to borderline.

---

### Decision · Program_Chairs · 2026-01-26

Reject